



**Measurement report: Dust impact on hygroscopicity and volatility of**
**submicron aerosols: Based on the observation in April of Beijing**
Xinyao Hu[1,2], Aoyuan Yu[1,3], Xiaojing Shen[1,2], Jiayuan Lu[1,4], Yangmei Zhang[1,2], Quan
Liu[1,2], Lei Liu[1,2], Linlin Liang[1,2], Hongfei Tong[1,2], Qianli Ma[5], Shuxian Zhang[5], Bing
Qi[6], Rongguang Du[6], Huizheng Che[1,2], Xiaoye Zhang[1,2], and Junying Sun[1,2*]
[1]State Key Laboratory of Severe Weather Meteorological Science and Technology, Chinese Academy
of Meteorological Sciences, Beijing, 100081, China
[2]Key Laboratory of Atmospheric Chemistry of CMA, Chinese Academy of Meteorological Sciences,
Beijing, 100081, China
[3]Dalian Meteorological Observatory, Dalian, 116000, China
[4]Department of Atmospheric and Oceanic Sciences and Institute of Atmospheric Sciences, Fudan
University, Shanghai, 200433, China
[5]Lin'an Atmosphere Background National Observation and Research Station, Lin'an 311307, Hangzhou,
China
[6]Hangzhou Meteorological Bureau, Hangzhou, 310051, China
*Correspondence to: Junying Sun (jysun@cma.gov.cn)
**Abstract.**
Understanding the aerosol hygroscopicity and volatility is crucial for determining their
effects on the environment and climate. As a typical natural aerosol, the dust impact on
fine particles' hygroscopicity and volatility remains inadequately understood.
Simultaneous measurements of aerosol hygroscopicity and volatility were performed
using Volatility-Hygroscopicity Tandem Differential Mobility Analyzer during April
2024 in Beijing. During this period, mean hygroscopic growth factor (HGF) of 50, 80,
110, 150, 200, and 300 nm were 1.20±0.07, 1.28±0.07, 1.32±0.07, 1.36±0.08,
1.40±0.09, and 1.43±0.13, respectively. The mean volatile shrink factor (VSF) was





0.48±0.05, 0.52±0.04, 0.53±0.05, 0.53±0.06, 0.53±0.07, and 0.54±0.10. Particles from
anthropogenic emissions were dominated by more hygroscopic and volatile
components, while particles influenced by natural sources (such as dust) had lower
hygroscopicity and volatility. The case study highlighted the impact of dust on
hygroscopicity and volatility for accumulated mode particles. Before dust arrival, more
hygroscopic and very volatile mode were more prominent, and HGF increased and VSF
decreased with diameter. When dust arrived, the number fraction of more hygroscopic
mode ($NF_{MH}$) dropped to 0.54 (200 nm) and 0.33 (300 nm), while number fraction of
very volatile mode ($NF_{VV}$) fell to 0.73 (200 nm) and 0.47 (300 nm), respectively. This
reflected a shift toward the hydrophobic and non-volatile components. During dust
period, the size dependence showed that HGF peaked at 150 nm and declined, whereas
VSF rose with diameter. The mean HGF and VSF at 300 nm were 1.20 and 0.74 during
dust period, suggesting that particles at 300 nm were hydrophobic and less volatile.

## 1. Introduction

Hygroscopicity and volatility are critical physical properties of atmospheric
aerosol particles. Hygroscopicity has a significant influence on atmospheric radiative
balance and visibility by altering particle size distribution and optical properties. In
addition, hygroscopicity indirectly affects the regional and global climate by
influencing the lifetime and microphysical properties of clouds (Gunthe et al., 2009;
Rose et al., 2010; Pöhlker et al., 2023). Volatility plays a crucial role in gas-particle
partitioning and the aging process of aerosols (Huffman et al., 2008). Considering the
hygroscopicity and volatility of aerosols in the model is of great significance for
reducing discrepancies between simulation results and observational data, and
improving the accuracy of model outputs (Gao et al., 2024; Mcfiggans et al., 2006;
Pringle et al., 2010; Rissler et al., 2010). Besides, determining the variation of particle
size at selected dry diameters under different relative humidity and temperatures can
also provide valuable indirect in-situ information regarding the chemical composition,
mixing state, and coating properties of aerosols (Massoli et al., 2010; Chen et al., 2022a;





Liu et al., 2025).
Given the significant environmental and climatic effects of aerosol volatility and
hygroscopicity, numerous field observations on aerosol volatility or hygroscopicity
have been conducted. Previous researches show that the volatility and hygroscopicity
of fine particles exhibit substantial differences due to the influence of emission sources
and atmospheric processes. The observed results in five sites across China by Chen et
al. (2022b) demonstrated that aerosols at suburban sites were more hygroscopic than
those in megacities, which may be attributed to the fact that suburban aerosols were
mainly from regional transport and thus more aged and well mixed. Cai et al. (2017)
found that aerosols exhibited lower volatility in Guangzhou compared to Okinawa,
because the aerosols in Guangzhou were affected more by traffic-related sources and
industrial emissions.
The size dependence of hygroscopicity and volatility is complex. Wu et al. (2016)
demonstrated a clear increasing trend in hygroscopicity with particle size, which was
similar to the size dependency of inorganic mass fraction in $PM_1$. While Shi et al. (2022)
found that aerosol hygroscopicity for 60, 100, 150, and 200 nm at 90% RH in rural
North China decreases with increasing particle size in winter, possibly due to enhanced
domestic heating emissions of non-hygroscopic or low-hygroscopicity primary aerosols
under low temperatures. In terms of volatility, the VSF of ambient aerosols in urban
Beijing decreases with increasing particle size (Wang et al., 2017), while Levy et al.
(2014) reported the opposite size dependence near the California-Mexico border.
Besides, the size dependence of volatility various under different pollution conditions.
Yu et al. (2025) found that volatility declined slightly with particle size under the clean
conditions, without an obvious size dependence under the pollution conditions.
Dust particles, suspended in the atmosphere, range from less than 0.1μm to over
100μm (Adebiyi et al., 2023). As one of the most important natural aerosols in the
atmosphere, dust aerosols significantly affect atmospheric chemistry, human health,
climate change, and biogeochemical cycles (Chen et al., 2021; Kurai et al., 2014; Lian
et al., 2025). Heterogeneous reactions between mineral dust and trace gases can alter



the chemical and physical properties of aerosols (Tang et al., 2017; Xu et al., 2020; Kok et al., 2023). Although the climatic and environmental effects of dust are considerable, limited studies focus on the dust effect on aerosol hygroscopicity and volatility simultaneously, especially on submicron aerosols (Kaaden et al., 2009; Kim and Park, 2012; Massling et al., 2007).

The North China Plain (NCP) is a hot spot of anthropogenic emissions, which can lead to severe air pollution. In recent years, the air quality in the NCP has significantly improved due to the strict control measures implemented by the government. However, the air pollution still happens due to unfavorable meteorological conditions, particularly in spring (Hu et al., 2021; Zhong et al., 2021). On the other hand, dust events often occur in the NCP in spring (Gui et al., 2023; Gui et al., 2022), which complicates the characteristics of aerosol hygroscopicity and volatility of the NCP. Although aerosol hygroscopicity and volatility in the North China Plain have been widely discussed in previous studies, existing work primarily focuses on the impact of chemical composition on aerosol hygroscopicity and volatility, the influence of anthropogenic emission on the hygroscopicity and volatility of aerosols, and the characterization of aerosol hygroscopicity and volatility under different pollution conditions(Wu et al., 2016; Chen et al., 2022a; Chen et al., 2022b; Shi et al., 2022; Zhang et al., 2023; Yu et al., 2025). The understanding of aerosol hygroscopicity and volatility in spring, particularly during the dust period, is limited.

In this study, the hygroscopicity and volatility of aerosols were measured simultaneously using a Volatility-Hygroscopicity Tandem Differential Mobility Analyzer (VH-TDMA) in the spring of 2024. The characteristics of aerosol hygroscopicity and volatility were analyzed, and the influence of air mass on aerosol hygroscopicity and volatility was discussed. Moreover, the relationships between hygroscopicity and volatility were explored. Finally, the impact of dust on the hygroscopicity and volatility of aerosol was investigated through a case study of a dust event. The study aimed to enhance understanding of aerosol hygroscopicity and volatility in spring, and to reveal the dust effect on the hygroscopicity and volatility of



fine particles.
**2. Experiment and instrumentation**
2.1 Sampling site
The measurements were performed at the Chinese Academy of Meteorological
Sciences (CAMS; 39.97°N, 116.37°E) from April 1st to April 29th, 2024. The CAMS
site is a typical urban site between the second-ring and third-ring roads in Beijing. There
is a major road within 200 meters to the west of the CAMS site. Local residents and
traffic emissions are the major sources of emissions at the CAMS site. More details
about the CAMS site are provided in the following studies(Zhang et al., 2023; Lu et al.,

121     2024).

2.2 Instrumentation
2.2.1 Aerosol hygroscopicity and volatility measurement
The size-resolved aerosol hygroscopicity and volatility were measured by a
Volatility Hygroscopicity Tandem Differential Mobility Analyzer (VH-TDMA)
(TROPOS, Germany), which is described in detail by Yu et al. (2025). Here, we just
introduced briefly. The sample air was dried by a silica dryer and a Nafion dryer to keep
its RH<30% before being pumped into the VH-TDMA. The VH-TDMA is composed
of an X-ray aerosol neutralizer (model 3088, TSI Inc., USA), two condensation particle
counters (CPC1 and CPC2; CPC 3772, TSI Inc., USA), and three medium Hauke-type
differential mobility analyzers (DMA1, DMA2, and DMA3, TROPOS, Germany),
custom-made Nafion dryers, thermal denuders (TDs), and humidifiers.
In this study, the VH-TDMA operates alternately in the H-TDMA mode and the
V-TDMA mode, which can measure the aerosol hygroscopicity and volatility
simultaneously. DMA1 selected quasi-monodisperse particles at selected diameters (Dp
= 50, 80, 110, 150, 200, and 300 nm). Then the aerosol particles of a certain size were
split into two flows. One flow passed through the TDs, followed by DMA2 and CPC1,



and the other flow passed through the humidifiers, and DMA3+CPC2. DMA2+CPC1
measured the size distribution of heated particles and DMA3+CPC2 measured the size
distribution of humidified particles, respectively. In this study, the temperature in the
TDs was set at 300 °C and the RH in the humidifier was set at 90%. In order to ensure
the accuracy of the RH measurement, the hygroscopicity of 100 nm ammonium sulfate
particles was measured regularly during the campaign. In this study, we used the
TDMAinv program package (Gysel et al., 2009) to inverted the data from VH-TDMA.
2.2.2 Particle number size distribution measurement
The particle number size distribution (PNSD) in the range of 8-850 nm under dry
conditions (RH<30%) was measured in spring of 2024 (including dust period) by a
tandem scanning mobility particle sizer (TSMPS, TROPOS, Germany). The TSMPS
consists of two differential mobility analyzers (DMAs, TROPOS, Germany) and two
condensation particle counters (CPCs, models 3772 and 3776, TSI Inc., St Paul, USA).
More information on PNSD measurement and setup is described in (Shen et al., 2018).
2.3 Data processing
The hygroscopic growth factor (HGF) is defined as the ratio of the particle's
electrical mobility diameter at a certain relative humidity to that under dry conditions:
$$\text{HGF} = \frac{Dp_{(wet)}}{Dp_{(dry,T0)}}$$
Where Dp(wet) is the particle diameter at an RH of 90%, Dp(dry,T0) refers to the
particle diameter at room temperature with an RH below 30%.
The volatile shrink factor (VSF) is defined as the ratio of the particle's electrical
mobility diameter at a certain temperature to that under dry conditions at room
temperature:
$$\text{VSF} = \frac{Dp_{(T1)}}{Dp_{(dry,T0)}}$$
Where Dp(T1) is the particle diameter at a set heating temperature of 300°C.
The hygroscopic growth factor probability density function (HGF-PDF) and
volatile shrink factor probability density function (VSF-PDF) were derived from the



number size distributions of humidified aerosol particles and volatile aerosol particles.
According to the HGF-PDF, aerosol particles were divided into two hygroscopic groups:
nearly hydrophobic (NH) mode particles (HGF ≤ 1.2) and more hygroscopic (MH)
mode particles (HGF > 1.2). Based on the VSF-PDF, aerosol particles were classified
into two groups: non-volatile (NV) mode particles (VSF ≥ 0.8) and very volatile (VV)
mode particles (VSF < 0.8). The detailed description of the calculation methods for
HGF-PDF, VSF-PDF, and the number fraction (NF) of each mode was shown in Yu et
al. (2025)
2.4 Other data used

Hourly $PM_{2.5}$ and $PM_{10}$ mass concentrations at the Guanyuan site were obtained

from the China National Environmental Monitoring Center. The meteorological data at
Haidian station (No.54399), which is located 5 km northwest of the CAMS site, was
obtained from the National Meteorological Information Center of the China
Meteorological Administration. Besides, the 48-hour back trajectories arriving at the
CAMS site were calculated using the HYSPLIT 4 model (Hybrid Single-Particle
Lagrangian Integrated Trajectory)(Draxler and Hess, 1998; Cohen et al., 2015). All data
in this study are reported in Beijing time (UTC+8).





**3. Results and discussion**
3.1 Overview

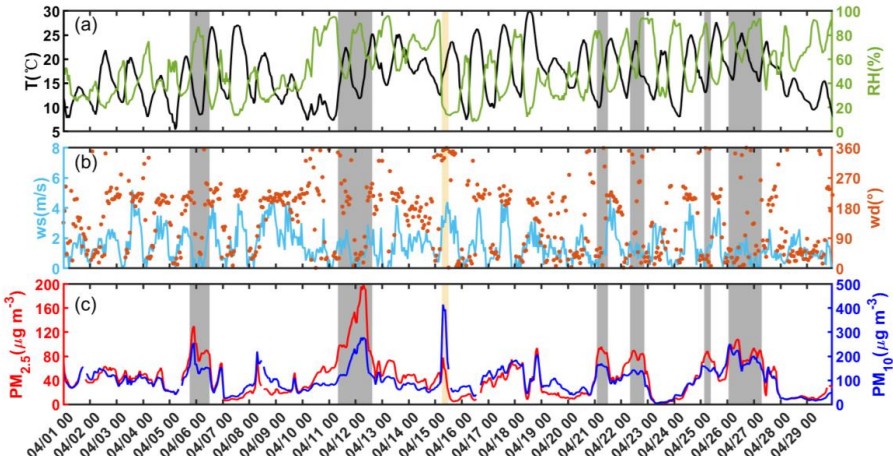


Figure 1. Time series of temperature and relative humidity (a), wind speed and wind
direction (b), $PM_{2.5}$ and $PM_{10}$ mass concentration (c) from April 1 to 29, 2024. The
gray-shade and yellow-shade area represent the pollution period and dust period,

respectively.

The time series of meteorological parameters and $PM_{2.5}$ and $PM_{10}$ mass

concentrations are depicted in Figure 1. During the study period, the average
temperature was 16.6±5.1 °C and the mean RH was 53.2±23.1%. The average wind
speed was 1.48 ±1.06 m/s, with the highest value up to 5.2 m/s. The mean $PM_{2.5}$ and
$PM_{10}$ mass concentrations were 44.9±32.3 and 100.7±57.0 $\mu g/m^3$ during the study
period. There were six pollution episodes and one dust episode during the whole
campaign. Here, the pollution episode was defined as the $PM_{2.5}$ mass concentration
exceeding 75 $\mu g/m^3$ and the ratio of $PM_{2.5}$ and $PM_{10}$ exceeding 0.3. Dust episode was
defined as the $PM_{10}$ mass concentration was larger than 100 $\mu g/m^3$ and the ratio of $PM_{2.5}$
and $PM_{10}$ was smaller than 0.3. During the pollution period, southerly winds prevailed
with low wind speed and high relative humidity. When dust arrived, the prevailing wind
is usually from the northwest with high wind speed with low RH.





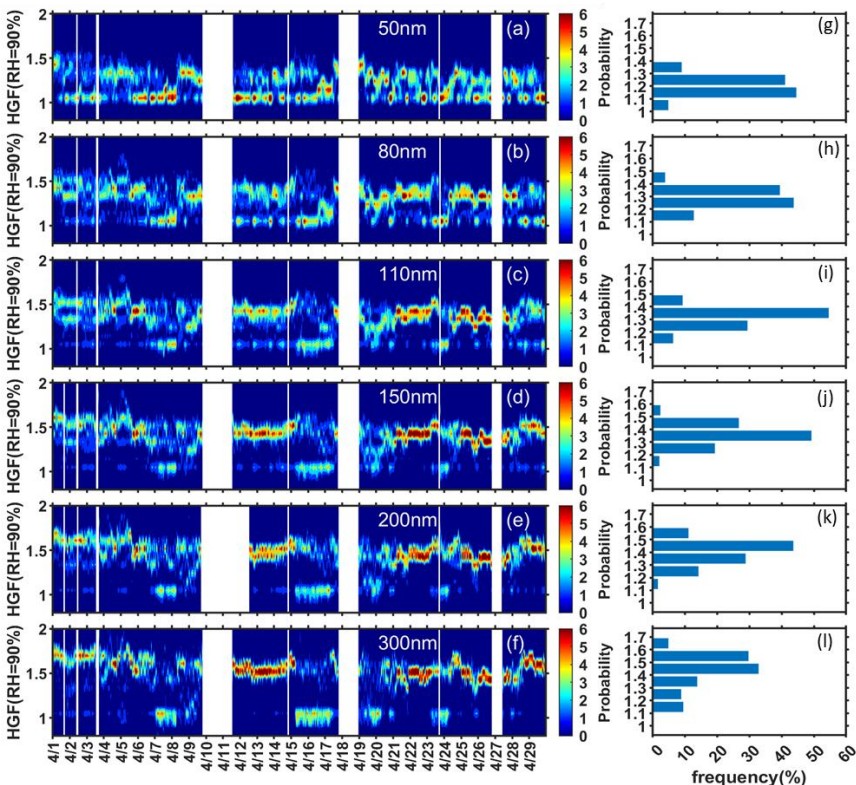

Figure 2. Time series of HGF-PDFs for 50, 80, 110, 150, 200 and 300 nm particles at 90% RH (a–f), and the frequency of the HGF values (g–l).

In this study the minimum HGF-PDF for 50, 80, 110, 150, 200 and 300 nm particles are typically at around 1.2 (Figure. S1), so the particles are divided into two modes: the nearly hydrophobic (NH) mode particles (HGF < 1.2) and more hygroscopic (MH) mode particles (HGF > 1.2). Figure 2 displays the time series of HGF-PDFs for 50, 80, 110, 150, 200 and 300 nm particles at 90% RH and the frequency of the HGF values. During the study period, the HGF for 50 nm particles was dominated by the NH mode, with the highest frequency of HGF ranging from 1.1 to 1.2. This could be related to the particles for 50nm, mainly influenced by local emission, such as traffic emissions. For 80, 110, 150, 200, and 300 nm particles, the MH mode in the HGF-PDF was basically more dominant. HGFs for 80 and 110 nm particles were mainly distributed between 1.2 and 1.3. For 150 and 200 nm particles, HGF mainly ranged from 1.3 to 1.4.



The distribution range of HGF for 300nm particles was wider, primarily ranging from
1.4 to 1.6. During the whole period, the number fraction of more hygroscopic mode
particles ($NF_{MH}$) for 50, 80, 110, 150, 200, and 300 nm particles was 51%, 68%, 75%,
79%, 80%, 77%, respectively. Only half of the particles at 50 nm were more
hygroscopic, which was due to the intensive emissions from traffic and cooking sources
around the site. The CAMS site is surrounded by residents and is near the main road
with heavy traffic. The particles from traffic and cooking sources are usually externally
mixed and hydrophobic(Henning et al., 2010). The NH mode was hardly observed in
aerosol particles ranging from 150 to 300nm during the pollution period, indicating the
particles were almost hygroscopic and more aged (Zhang et al., 2023). As shown in
Table 1, HGF shows a strong size dependency. The mean values of HGF for 50 - 300nm
particles were $1.20\pm0.07$, $1.28\pm0.07$, $1.32\pm0.07$, $1.36\pm0.08$, $1.40\pm0.09$, $1.43\pm$
0.13, which were consistent with the results observed by Wang et al. (2017) that aerosol
particles became more hygroscopic with increasing particle size.





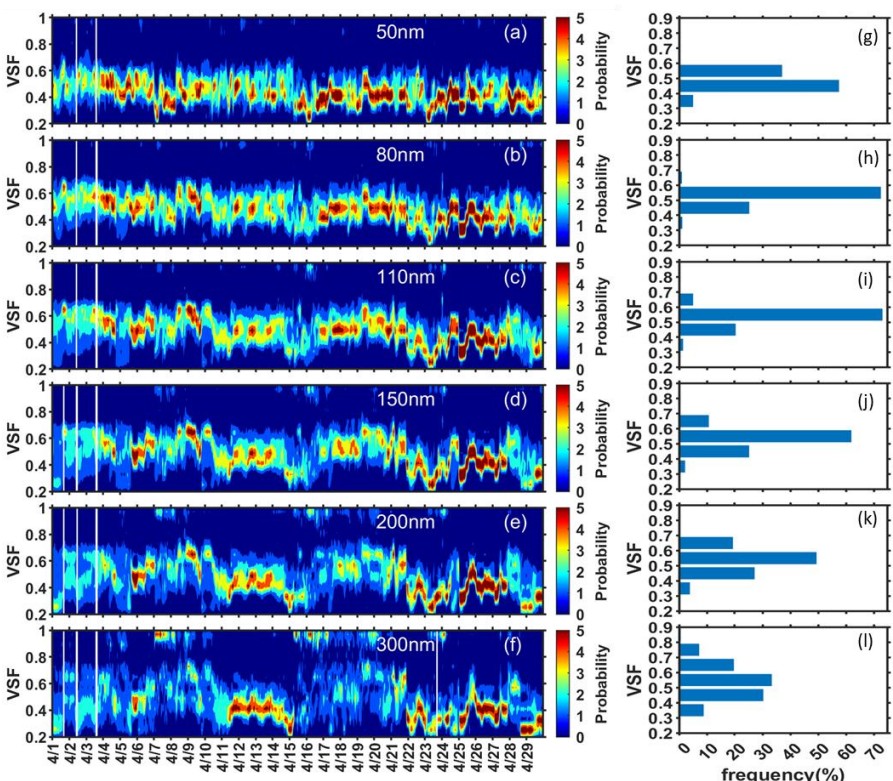


Figure 3. Time series of VSF-PDF for 50, 80, 110, 150, 200 and 300 nm particles (a–
f), and the frequency of the VSF values (g–l).
After the aerosol is heated to 300℃, the remaining substances are non-volatile
substances, such as elemental carbon and dust. As shown in Figure S1, the mean VSF-
PDFs show a minimum value at around 0.8, and the particles are separated into two
modes: the non-volatile (NV) mode particles (VSF > 0.8) and the very volatile (VV)
mode particles (VSF ≤ 0.8). Figure 3 shows the variation of volatile shrink factor
probability density functions (VSF-PDFs) and the frequency distribution of VSF for 50,
80, 110, 150, 200 and 300 nm particles. The VSF-PDFs for particles in the 50-300 nm
range were mainly dominated by the VV mode, with occasional enhancement of the
NV mode for particles at 200, and 300 nm. During the sampling period, the mean
number fraction of very volatile mode particles ($NF_{VV}$) for 50, 80, 110, 150, 200 and
300 nm particles was 96%, 95%, 94%, 93%, 91%, 87%, respectively, indicating the





particles were almost volatile. VSF for 50 -150nm particles was mainly concentrated in
the range between 0.4 and 0.6. The distribution range of VSF for 300nm particles was
wider, suggesting that the volatility of 300 nm aerosol particles varied greatly. During
the study period, the mean value of VSF for 50 - 300nm particles was $0.48\pm0.05$, $0.52$
$\pm0.04$, $0.53\pm0.05$, $0.53\pm0.06$, $0.53\pm0.07$, $0.54\pm0.10$, respectively.
Table 1. The hygroscopicity an volatility parameter during the campaign.

|  |  | 50nm | 80nm | 110nm | 150nm | 200nm | 300nm |
|---|---|---|---|---|---|---|---|
| Total | $NF_{MH}$ | 0.51±0.21 | 0.68±0.17 | 0.75±0.15 | 0.79±0.14 | 0.80±0.16 | 0.77±0.22 |
|  | HGF | 1.20±0.07 | 1.28±0.07 | 1.32±0.07 | 1.36±0.08 | 1.40±0.09 | 1.43±0.13 |
|  | $NF_{VV}$ | 0.96±0.04 | 0.95±0.04 | 0.94±0.05 | 0.93±0.05 | 0.91±0.07 | 0.87±0.12 |
|  | VSF | 0.48±0.05 | 0.52±0.04 | 0.53±0.05 | 0.53±0.06 | 0.53±0.07 | 0.54±0.10 |
| Dust | $NF_{MH}$ | 0.44±0.11 | 0.50±0.15 | 0.57±0.13 | 0.62±0.08 | 0.54±0.08 | 0.33±0.15 |
|  | HGF | 1.20±0.04 | 1.26±0.07 | 1.30±0.06 | 1.34±0.05 | 1.31±0.05 | 1.20±0.08 |
|  | $NF_{VV}$ | 0.93±0.05 | 0.89±0.04 | 0.87±0.04 | 0.83±0.04 | 0.73±0.09 | 0.47±0.21 |
|  | VSF | 0.46±0.04 | 0.51±0.02 | 0.52±0.02 | 0.53±0.02 | 0.57±0.03 | 0.74±0.12 |

3.2 Influence of air mass on aerosol hygroscopicity and volatility
The hygroscopicity and volatility of aerosols are closely related to the source of
air masses (Cai et al., 2017). The chemical components, aging degrees, and mixing
states of aerosols carried by air masses originating from different regions are different,
which leads to significant differences in the properties of aerosols corresponding to
different air masses. In order to investigate the influence of air mass on aerosol
hygroscopicity and volatility. The 48-h back trajectories were calculated, and the back
trajectories were classified into four clusters. As shown in Figure 4, cluster 1 originated
from the Bohai Sea, passed through Tianjin, and reached Beijing, which accounted for
27% of the total back trajectories. Cluster 2 originated from Mongolia, and had the
longest transport distance. Cluster 2 arrived in Beijing through Inner Mongolia and the
northwest of Hebei province. Cluster 3, which accounted for 39% of total back
trajectories, arrived in Beijing from the Shanxi province and southwest of Hebei
province. Cluster 4 originated from the northeast of Inner Mongolia and passed through





the Liaoning Province and the northeast of Hebei Province. The HGF corresponding to
Cluster 1 was significantly greater than that of the other clusters. This might be related
to the fact that Cluster 1 passed through the Bohai Sea during its transport, carrying
more hygroscopic marine aerosols. The HGF of cluster 1 at 150 nm is 1.41, which is
close to the result of ambient marine aerosol particles at 145nm (Hakala et al., 2016).
Cluster 2, which originated from the northwest, had a smaller HGF and a larger VSF,
suggesting that the particles from the northwestern air mass had weak hygroscopicity
and weak volatility. This is because Cluster 2 passed through the Gobi Desert during its
transport, carrying dust aerosols, which led to a decrease in the aerosol hygroscopicity
and volatility.

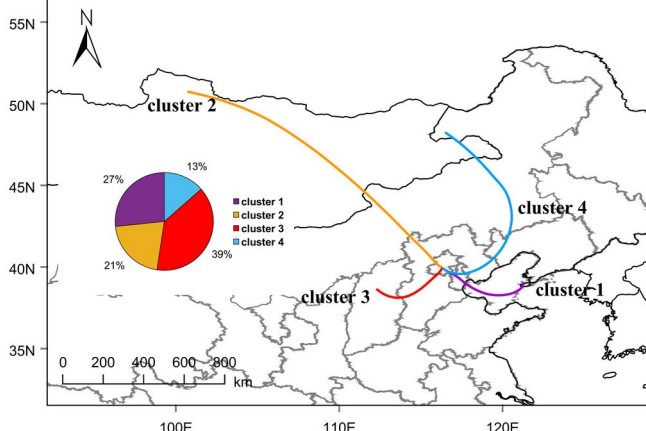


Figure 4. Air mass clusters of 48-h back trajectories arriving at the CAMS site in
Beijing during the study period, and the fraction of each cluster accounting for the
total back trajectories.



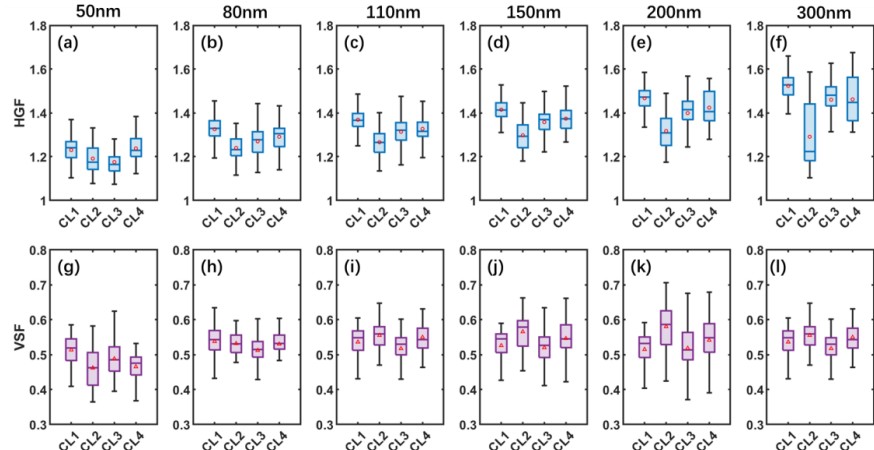

Figure 5. The variation of size- resolved HGF (a-f) and VSF (g-l) in each cluster in the campaign. The solid line in the box represents the median value, and the dot indicates the mean value. The box contains the range of values from 25 % (bottom) to 75 % (top), and the upper and lower whiskers are the 95th and 5th percentiles, respectively.





3.3 Relationship between aerosol hygroscopicity and volatility

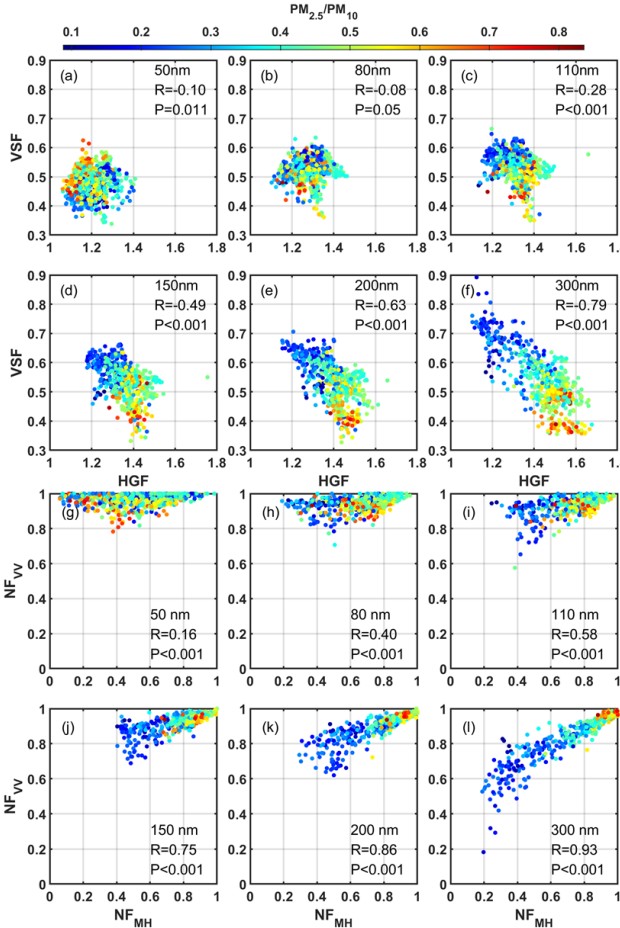

Figure 6. The relationship between HGF and VSF (a-f), as well as $NF_{VV}$ and $NF_{MH}$ (g-l). Color bar represents the mass ratio of $PM_{2.5}$ and $PM_{10}$.

In order to explore the relationships between aerosol hygroscopicity and volatility, the relationships between HGF and VSF, as well as $NF_{VV}$ and $NF_{MH}$, were analyzed. Figure 6 exhibits the relationship between HGF and VSF, as well as $NF_{VV}$ and $NF_{MH}$. During the observation period, there was no significant correlation between the HGF and VSF of the 50nm and 80nm aerosol particles, and the correlations between the $NF_{VV}$ and $NF_{MH}$ of the 50nm and 80nm aerosol particles were relatively weak. Previous studies in Beijing also found similar results, which might be related to the low degree of ageing in Aitken mode particles (Wang et al., 2017). The accumulation mode



particles exhibited a significant negative correlation between HGF and VSF, and the
correlation became strong with increasing particle size. The relationship between
hygroscopicity and volatility of accumulation mode particles was different from that of
Aitken mode particles, which could be related to the aerosol chemical composition.
Previous studies in Beijing revealed that the particles below 100nm are mainly organics
(Li et al., 2023), while accumulation mode particles are dominated by secondary
aerosols (Xu et al., 2015).

The color bar in Figure 6 represents the ratio of $PM_{2.5}$ to $PM_{10}$, which is used to

distinguish anthropogenic pollution from dust episodes (Querol et al., 2001). A higher
$PM_{2.5}/PM_{10}$ ratio usually indicates that fine particulate matter and secondary particles
are the main contributors, while a lower $PM_{2.5}/PM_{10}$ ratio indicates that coarse
particulate matter generated by natural processes (such as dust) is dominant (Zha et al.,
2021; Li et al., 2020). As shown in Figure 6, when the mass ratio of $PM_{2.5}$ to $PM_{10}$ was
less than 0.3, the $NF_{MH}$ and $NF_{VV}$ of accumulation mode particles were low. This
indicated that lower hygroscopicity and lower volatility components dominated
accumulation mode aerosols when particles from natural sources were dominant in the
atmosphere. However, when the ratio of $PM_{2.5}$ to $PM_{10}$ was larger than 0.3, the $NF_{MH}$
and $NF_{VV}$ of accumulation mode particles were larger. This suggested that when fine
particles from anthropogenic emissions were dominant in the atmosphere, the
accumulation mode particles primarily consisted of higher-hygroscopicity and higher-
volatility components. Besides, when the mass ratio of $PM_{2.5}$ to $PM_{10}$ was less than 0.3,
the particles for 150-300 nm exhibited a low HGF and high VSF, while when the mass
ratio of $PM_{2.5}$ to $PM_{10}$ exceed 0.3, the particles for 150-300 nm showed a high HGF
and low VSF. This suggested that the hygroscopicity and volatility of accumulated
mode aerosols from natural or anthropogenic sources were different. The discrepancy
in the values of HGF and VSF at 300nm under different mass ratios of $PM_{2.5}$ to $PM_{10}$
was more pronounced, indicating that anthropogenic and natural sources had different
effects on accumulation mode particles of varying sizes.





3.4 The evolution of aerosol hygroscopicity and volatility during dust period

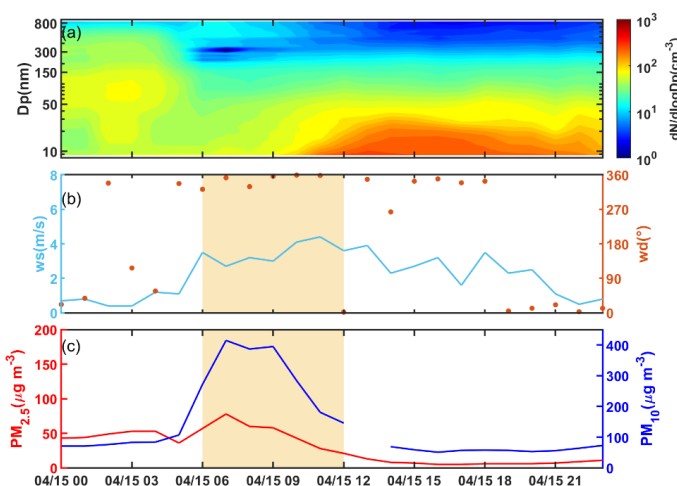

Figure 7. The variation of particle number size distribution, wind speed and wind
direction, PM$_{2.5}$ and PM$_{10}$ mass concentration on April 15, 2024. The yellow-shade
area represents the dust period.

During the study period, a dust episode was observed on April 15, 2024. In

response to this dust event, the Beijing Meteorological Observatory issued a blue dust
warning (https://yjglj.beijing.gov.cn/art/2024/4/15/art_2472_674730.html, last access:
July 30, 2025). Before 06:00 on April 15, the wind speed was relatively low, with PM$_{2.5}$
levels approximately at 50 μg/m$^3$. The ratio of PM$_{2.5}$ to PM$_{10}$ was high (~0.6), which
indicated that fine particles from anthropogenic emissions dominated. The period from
0:00 to 5:00 on April 15 was defined as the before dust period (before-DS). At 6:00, the
wind speed increased, and the wind direction was predominantly North. PM$_{2.5}$ and PM$_{10}$
mass concentrations rose rapidly in response to high wind speed, and reached up to 78
and 415 μg/m$^3$, respectively. Then the PM$_{2.5}$ and PM$_{10}$ mass concentrations decreased
gradually. The average ratio of PM$_{2.5}$ to PM$_{10}$ between 6:00 and 12:00 was 0.16,
indicating that the dust particles were dominant. The backward trajectory shows that
the air mass between 6:00 and 12:00 mainly originated from the central and western of
Mongolia, passing through Inner Mongolia and Hebei province before reaching Beijing.
So, the dust period (DS) was defined between 06:00 and 12:00 on April 15. After 12 o



'clock, the mass concentrations of PM$_{10}$ was lower than 100 µg/m$^3$, and the wind speed

also gradually decreased. So, the after-dust period (after-DS) was defined between

13:00 and 23:00 on April 15. Besides, a new particle formation (NPF) event occurred

during this period, which is the case with the appearance of the nucleation mode, but

without clear growth. As shown in Figure S2, after 12:00, the back trajectories still

originated from the north of Beijing and passed through Mongolia, Inner Mongolia,

which may still bring the dust particles.

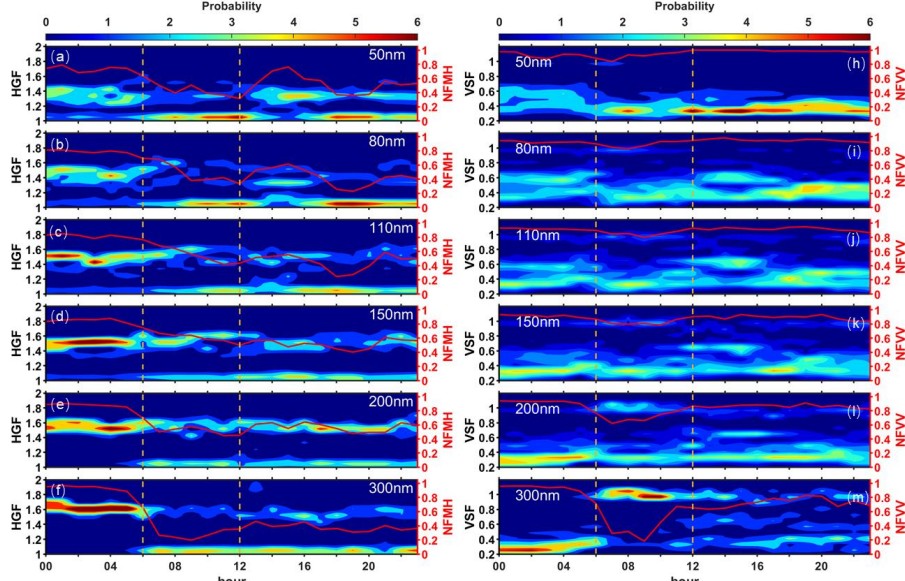

Figure 8. The variation of HGF-PDFs and VSF-PDFs for 50, 80, 110, 150, 200 and

300 nm particles on April 15, 2024. The area between the yellow dashed lines

indicates the dust period.



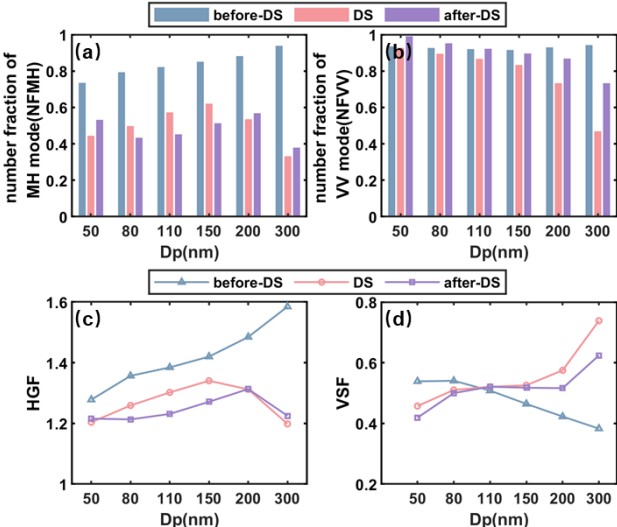


Figure 9. The number fraction of MH mode and VV mode, the mean value of HGF
and VSF for 50-300 nm on April 15, 2024.
Figure 8 illustrates the variation of HGF-PDFs and VSF-PDFs for 50, 80, 110, 150,
200, and 300 nm particles on April 15, 2024. Before the dust period, the more
hygroscopic mode was more prominent, no matter what particle size was considered.
With increasing particle size, the dominance of the more hydroscopic mode became
more pronounced. The number fraction of MH mode for 300nm particles was as high
as 93.9%. In terms of volatility, the very volatile mode particles for 50-300nm particles
were dominant, with the number fraction of VV mode exceeding 80%. As shown in
Figure 9, both HGF and VSF were strongly size-dependent before the dust period. As
the increase of particle size, HGF increased and VSF decreased, indicating that the
volatility and hygroscopicity of smaller aerosols were weaker than those of larger
aerosol particles. A similar size dependence of HGF and VSF was also observed at the
California-Mexico border (Levy et al., 2014).
During the dust period, the proportion of more hygroscopic mode particles
decreased rapidly. When the dust arrived, the strong north wind not only brought dust
particles but also swept pre-existing fine particulate matter in the atmosphere. The most
significant decline in the number fraction of MH mode particles was observed at 200



and 300nm. The mean value of the number fraction of MH mode particles for 200 and
300nm was 0.54 and 0.33, which were much lower than that before the dust period.
Previous research also found that the number fraction of MH mode particles for 250
and 350 nm was less than 0.5 during the dust event (Kaaden et al., 2009).
Simultaneously, the NV mode of 200 and 300 nm became more prominent. During the
dust period, the minimum $NF_{VV}$ was only 0.18. The mean value of the number fraction
of VV mode for 200 and 300 nm particles decreased from 0.93 and 0.94 before the dust
period to 0.73 and 0.47 during the dust period, respectively. The decrease in the number
fraction of MH mode and the number fraction of VV mode particles for 200 and 300
nm reflected a shift toward hydrophobic and non-volatile components. Besides, HGF
and VSF exhibited different size dependence during the dust period compared with
before the dust. During the dust period, HGF reached a peak at 150nm and then declined
as the particle size increased (Figure 9c). The mean HGF for 300nm was 1.20, which
was close to the observed results of Massling et al. (2007) during the dust period.
During the dust period, the VSF for 50nm was the least, and the VSFs for 80-150 nm
were close. While VSF increased with the increase of particle size from 150-300 nm,
reaching as high as 0.74 at 300nm during the dust period (Figure 9d).

During the after-dust period, the number fraction of MH mode particles for 50-

300 nm gradually increased, but remained below that before the dust period.
Particularly, the MH modes of 50 nm became strong and the number of MH mode of
50nm gradually increased and reached 0.76 at 3:00 (Figure. 8a). This might be related
to the NPF event that occurred after the dust. Zhang et al. (2023) reported that the
hygroscopicity of 50 nm particles was usually affected by NPF events. Previous studies
demonstrated that a significant enhancement in the hygroscopicity of 40nm organic
aerosol particles during NPF(Liu et al., 2021) and an increase in the proportion of water-
soluble compounds in newly formed particles(Shantz et al., 2012; Wu et al., 2016).
During the evening rush hour, it can be clearly observed that the number fraction of
MH mode for 50-110 nm particles decreased markedly, probably owing to fresh traffic
and cooking emissions during the evening rush hour (Figure 8a-c). After the dust, the



VV mode of 50nm particles became more pronounced (Figure 8g), which could be
related to the volatile matters that produced during the NPF event (Wu et al., 2017).
The average VSF at 50nm was 0.41 after the dust, which was close to the VSF at 50nm
observed during the NPF event in Wu et al. (2017). Size-dependent variations in HGF
and VSF during the after-dust period resembled those during the dust period. After the
dust, HGF at 200 and 300nm were 1.31 and 1.22. The VSFs at 200 and 300nm were
0.52 and 0.62, which were lower than those during the dust period.

## 4. Conclusion

Simultaneous measurements of hygroscopicity and volatility of aerosol were
performed using a Volatility-Hygroscopicity Tandem Differential Mobility Analyzer
(VH-TDMA) in Beijing during April 2024. Results show that 50 nm particles were
dominated by the nearly hydrophobic (NH) mode, which could be related to the
particles for 50nm being mainly influenced by local emissions, such as traffic emissions
and cooking sources. For 80, 110, 150, 200, and 300 nm particles, the more hygroscopic
(MH) mode in the HGF-PDF was basically more dominant. During the study period,
the mean HGF values for particles ranging from 50 to 300 nm were 1.20±0.07,
1.28±0.07, 1.32±0.07, 1.36±0.08, 1.40±0.09, and 1.43±0.13, respectively. For 50–300
nm particles, the VSF-PDF was mainly dominated by the very volatile (VV) mode, with
mean VSF values during the study period being 0.48±0.05, 0.52±0.04, 0.53±0.05,
0.53±0.06, 0.53±0.07, and 0.54±0.10 for the respective sizes.
Back trajectory analyses indicated that Cluster 2, originating from the northwest,
exhibited a smaller HGF and a larger VSF, suggesting that the particles from the
northwestern air mass had weak hygroscopicity and weak volatility. This could be due
to the fact that Cluster 2 passed through the Gobi Desert during its transport, carrying
dust aerosols, which led to a decrease in the aerosol hygroscopicity and volatility. The
relationships between the hygroscopicity and volatility of aerosol show that the
accumulation mode particles exhibited a significant negative correlation between HGF
and VSF, and the correlation became strong with an increase in size. Besides, when the





mass ratio of $PM_{2.5}$ to $PM_{10}$ was less than 0.3, the $NF_{MH}$ and $NF_{VV}$ of accumulated
mode particles were low, suggesting that accumulated mode particles were dominated
by components with lower hygroscopicity and volatility when particles from natural
sources were dominant in the atmosphere. Conversely, when the dominance of fine
particles from anthropogenic emissions ($PM_{2.5}/PM_{10}$ >0.3), higher $NF_{MH}$ and $NF_{VV}$
values for accumulation mode particles indicated that they were primarily composed of
more hygroscopic and volatile components.

The influence of dust process on the hygroscopicity and volatility of aerosol at

different sizes was also investigated. Before the dust period, the more hygroscopic
mode was prominent regardless of size. The number fraction of MH mode increased
with particle diameter (93.9 % at 300 nm). As for volatility, very volatile (VV) mode
was dominant and the number fraction of VV mode for 50-300nm particles was higher
than 80%. Both HGF and VSF exhibited a clear size dependence: as the increase of
particle size, HGF increased and VSF decreased, indicating that smaller particles were
less hygroscopic and less volatile. Compared with before the dust period, the mean MH
mode fraction dropped sharply to 0.54 for 200 nm and 0.33 for 300 nm, and the mean
VV mode fraction decreased to 0.73 (200 nm) and 0.47 (300 nm) during the dust period.
This reflected a shift toward hydrophobic and non-volatile components. During the dust
period, HGF reached a peak at 150nm and then declined as the particle size increased.
The mean HGF was 1.20 for 300 nm during the dust period. While VSF increased with
the increase of particle size from 150 to 300 nm, reaching as high as 0.74 at 300nm.
After the dust, the MH mode fraction gradually increased but remained below that
before the dust period. An NPF event temporarily elevated the MH mode fraction of 50
nm particles to 0.76, while fresh traffic and cooking emissions reduced it during the
evening rush hour. Affected by the NPF event, the VV mode of 50 nm particles was
enhanced. After the dust, the size-dependent patterns of HGF and VSF resembled those
during the dust period, demonstrating a persistent dust influence. To our knowledge,
this work focuses on the dust impact on the hygroscopicity and volatility of fine
particles simultaneously in the North China Plain for the first time. The results reveal



that the hygroscopicity and volatility of accumulated mode particles are significantly
influenced by dust. Accumulated mode plays a significant role in cloud condensation
nuclei (CCN) activation. Further research is needed to quantify the changes in
accumulation mode properties influenced by dust, specifically their impact on CCN
activation.
**Data availability.**
The full data are available at https://doi.org/10.5281/zenodo.16957115 (Hu et al., 2025).
**Competing interests.**
The authors declare that none of the authors has any competing interests.
**Author contributions.**
XH analyzed the observational data, prepared the figures and wrote the original draft.
AY conducted the instrument deployment, measurement and data processing. JS
designed the experiment, reviewed and finalized the manuscript. All co-authors
discussed the results and commented on the manuscript.
**Acknowledgements.**
This research was supported by the Innovation Team for Haze-fog Observation and
Forecasts of MOST.
**Financial support.**
This  study  was  supported  by  the  China  Meteorological  Administration
(CXFZ2024J039), the National Key Research and Development Program of China
(2023YFC3706305, 2024YFC3712902), the National Natural Science Foundation of
China (42475121, 42275121,42075082), Basic Research Fund of CAMS (2025Y002,
2023Z012, 2024Z006). It was also supported by the Innovation Team for Haze-fog
Observation and Forecasts of MOST.



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
