# Peer review of "Measurement report: Dust impact on hygroscopicity and volatility of"

_EGUsphere, 2025_

## Author Comment (AC1)

**Response to Reviewer #1**

Review of "Measurement report: Dust impact on hygroscopicity and volatility of submicron aerosols: Based on the observation in April of Beijing" by Hu et al. This manuscript investigates the hygroscopicity and volatility of submicron aerosols during a one-month campaign in urban Beijing. The authors present temporal variability in relation to meteorological parameters and $PM_{2.5}$/$PM_{10}$ mass concentrations, provide a statistical overview of the campaign, analyze air-mass back trajectories, and examine the impact of air mass on aerosol properties. Particular attention is given to a short dust event, with a comparison of aerosol hygroscopicity and volatility before, during, and after the episode. The topic is relevant and of potential interest to the community, particularly with respect to understanding the role of dust in influencing aerosol hygroscopic properties. However, I find that the current manuscript has several limitations that, in its present form, raise concerns regarding its suitability for publication in ACP. Below I provide detailed comments and suggestions for improvement:

**1. Overall approach and novelty**

The manuscript is largely descriptive, particularly in Sections 3.1 and 3.2. The discussion would benefit from a stronger connection to existing literature, as the current version does not sufficiently highlight what is novel compared to earlier studies from the region. I encourage the authors to clarify the new insights gained from this dataset and to better emphasize the scientific significance of their results.

**Reply:** Thanks for reviewer's suggestion. To address your concern, we strengthened the discussion in the manuscript, especially in Sections 3.1 and 3.2, and we reorganized the manuscript to clarify the new insights and the scientific significance of the results in the manuscript.

**2. Scope and dataset**

The dataset, consisting of one month of HV-TDMA measurements without complementary observations of aerosol chemical composition or other physical/optical properties, appears rather limited. As presented, the analysis remains descriptive, which may not meet the standards expected for ACP. The authors may consider expanding the

contextualization of their results within the broader body of work on aerosol hygroscopicity and volatility to strengthen the manuscript.

**Reply:** Thanks for your comments. In order to improve the manuscript, we have added a discussion on the optical properties of aerosols in the case study. We added a comparative discussion regarding the research results of this study and those of previous studies. We have revised the manuscript carefully according to reviewers' suggestions and comments, hopefully, it is suitable to the scope of the ACP.

**3. Title and focus on dust event**

The title suggests a comprehensive assessment of dust impacts; however, the analysis relies on a single, short-lived dust event (approximately 6 hours). Drawing broad conclusions on dust impacts from such limited data seems overly ambitious. A more cautious framing of the study in the title and conclusions would be appropriate. Ideally, a longer dataset including multiple dust events would allow for more robust evaluation of dust effects on aerosol properties.

**Reply:** Thanks for reviewer's suggestion. We agree that a longer dataset including multiple dust events is essential for drawing definitive conclusions on dust impacts. Unfortunately, there was only one dust events during the study period. To address this issue, we have modified the manuscript title, reorganized the content, and explicitly noted the limitation of the short-duration dataset in the conclusion. Besides, we have also discussed the hygroscopicity enhancement factor based on aerosol scattering coefficient measurements to enhance the assessment of dust impacts. Please see more details in Line 475-497 in the revised manuscript. We plan to conduct long-term measurements to obtain more data on dust events in the future research to validate and extend the current findings, which will enable a more robust evaluation of dust effects on aerosol properties as recommended. We modified the sentence as:

"*This study enhances understanding aerosol mixing state and evolution under different conditions and offer reliable observational constraints for reducing discrepancies between simulation results and observational data. Although only one dust event was captured, the results offer valuable insights into the impact of dust on accumulation mode particles. In the future, more observations including aerosol hygroscopicity, volatility and*

*chemical composition are needed during dust period to better quantify the influence of dust on aerosol properties.*" (Line 592-598 in the revised manuscript)

**4. Characterization of the dust event**

Additional supporting information would help establish the identification of this episode as a dust event. How exactly is the event defined? While changes in the fine/coarse ratio are a useful indicator, dust events (especially those associated with long-range transport) typically last longer than a few hours. It would strengthen the analysis to incorporate additional observational evidence (e.g., satellite data, lidar, or ground-based measurements) if available. I also note that the link provided to the Beijing Meteorological Observatory does not appear to work and should be updated or replaced.

**Reply:** Thanks for reviewer's suggestion. We fully agree that additional supporting evidence is essential to robustly confirm the nature of dust event. To clearly define the haze pollution and dust event, we added a subsection in the Methods section as follow:

"*In Beijing, haze pollution (hereinafter referred to as EP) and dust pollution (hereinafter referred to as DS) frequently occur in the spring (Liu et al., 2023). Episodes of EP are typically characterized by a significant increase in PM mass concentration, accompanied by a high $PM_{2.5}/PM_{10}$ ratio, whereas DS events are associated with a sudden rise in PM mass concentration but a low $PM_{2.5}/PM_{10}$ ratio (Wang et al., 2019a; Fu et al., 2010; Wang et al., 2015; Tong et al., 2017). In this study, EP was defined based on the Class II standard values of the Chinese Ambient Air Quality Standards, as well as the $PM_{2.5}/PM_{10}$ ratio. Specifically, an EP event was defined when the $PM_{2.5}$ mass concentration exceeding 75 $\mu g/m^3$ and the ratio of $PM_{2.5}$ and $PM_{10}$ was greater than 0.3. During the study period, a total of six haze pollution (EP) episodes were identified. For DS events, the identification process involved two main steps. The first step identifying the DS event was to determine the particle mass concentration. We took the $PM_{10}$ mass concentration was larger than 100 $\mu g/m^3$ and the ratio of $PM_{2.5}$ and $PM_{10}$ was smaller than 0.3 as the threshold to select possible DS events(Wang et al., 2019a; Wang et al., 2015). Second, the Atmospheric Environmental Meteorological Bulletin 2024 was consulted to confirm the occurrence of reported dust events, including their primary affected areas. Based on this combined approach, one DS event that occurred in Beijing in April 2024 was identified.*" (Line 223-241 in the revised

Besides, we added the discussion about the variation of aerosol optical properties to explore the dust event.

"*To further discuss the changes in aerosol properties during dust period, we analyzed the variations in aerosol optical properties. Figure 9 shows the time series of scattering coefficient, scattering Angstrom exponent, aerosol scattering hygroscopic growth factor for $PM_1$ and $PM_{10}$ on April 15, 2024. The mean values of $\sigma_{sp}$ for $PM_1$ and $PM_{10}$ were 100.9 and 259.8 $Mm^{-1}$ during the dust period. The averages of SAE for $PM_1$ and $PM_{10}$ decreased sharply from 1.69 and 1.27 before dust period to 0.02 and -0.16 during the dust period, suggested a significant shift in aerosol size distribution toward coarse particle dominance (Hu et al., 2021). The persistently low SAE for $PM_1$ and $PM_{10}$ values throughout the dust episode suggested the larger particles were dominated. . During the dust period, SAE was comparable to previous results during the dust periods in Beijing and Nanjing (Song et al., 2023; Xia et al., 2019). The mean f(80%) for $PM_1$ and $PM_{10}$ decreased from 1.62 and 1.60 before the dust to 1.03 and 1.02 during the dust period, suggested that the submicron and super micron aerosols are almost hydrophobic during the dust period. The f(80%) during the dust period was similar to the value that observed in Beijing dust period (Xia et al., 2019). Although the scattering coefficient for $PM_1$ and $PM_{10}$ still remained a low level after the dust period, SAE and f(80%) gradually rose, implying that the fine particles from anthropogenic sources gradually became dominant.*" (Line475-497 in the revised manuscript)

We have re-checked the link to the Beijing Meteorological Observatory thoroughly through multiple browsers (Chrome, Firefox, and Edge) and network environments, and confirmed that it is accessible. The link website screenshot as follow:

[Figure]

**Recommendations for improvement**

Rather than centering the study on the short dust episode, I recommend the authors consider broadening the scope of their analysis to other features of the campaign dataset. For example, the six identified pollution episodes may provide a stronger basis for discussion and allow for more meaningful conclusions regarding aerosol hygroscopicity and volatility. Additionally, engaging more deeply with the existing literature would improve the scientific context and highlight the added value of this dataset.

**Reply:** Thank you for your constructive suggestions. We have restructured the manuscript and investigated of aerosol hygroscopicity and volatility under different environment and compared our results with previous studies. Please see more details in section 3.4 (Line 407-453 in the revised manuscript). We also showed as below:

*"During the study period, significant differences were observed in the hygroscopicity and volatility of aerosols under different conditions. As shown in Figure 7, with increasing particle size, the dominance of the more hygroscopic mode became more pronounced under haze pollution conditions, implying that the larger size particles exhibited higher hygroscopicity under haze pollution conditions. The number fraction of $NF_{MH}$ at 200 and 300 nm was close to 1, indicating that the particles at 200 and 300 nm were almost internally mixed and highly aged (Wang et al., 2019b). In terms of volatility, the VV mode particles were predominant for 50-300 nm particles, with the number fraction of VV mode particles exceeding 0.9 regardless of particle size.*

*NFvv for both 50 nm and 300 nm particles was close to 1 under haze pollution conditions, whereas $NF_{MH}$ for 50 nm and 300 nm particles was 0.39 and 0.94, respectively. These distinct hygroscopic and volatile characteristics of 50 nm and 300 nm particles under haze conditions may be associated with differences in their sources and aging processes. Previous study found that the dominant sources of 40 nm particles under polluted conditions were emissions from cooking and traffic (Wang et al., 2019b). Freshly emitted organic aerosols from vehicles are characterized by low-oxidation state, high volatility, and low hygroscopicity (Tiitta et al., 2010; Feng et al., 2023). A study conducted in Beijing reported that cooking organic aerosols (COA) plays a critical role in modifying aerosol hygroscopicity, with higher COA fractions leading to decreased hygroscopicity of organic aerosols (Liu et al., 2021). Additionally, heterogeneous reactions (e.g., aqueous-phase reactions) have been demonstrated to considerably affect the hygroscopicity of accumulation mode particles during haze episodes (Wang et al., 2019b). During haze pollution periods, reduced solar radiation and persistently high relative humidity promote the formation of secondary inorganic components through liquid-phase and heterogeneous reactions, accelerating particle aging in polluted environments (Sun et al., 2016). As a result, internally mixed accumulation mode particles become abundant and exhibit high hygroscopicity and volatility during haze pollution (Wu et al., 2016; Wang et al., 2019b). The HGF increased significantly with particle size under haze conditions, consistent with previous findings (Zhang et al., 2023a), likely due to the higher mass fraction of secondary inorganic aerosols in larger particles (Wang et al., 2018a). VSF did not show a markable size dependency, which is similar to the result under the pollution period observed by Wang et al. (2017) during haze episodes.*

*During the dust period, $NF_{MH}$ initially increased with particle size but subsequently declined, while NFvv gradually decreased. These suggested that a high proportion of particles with low hygroscopicity and low volatility at 200 and 300 nm. Both HGF and VSF displayed distinct size-dependent behavior during dust events compared to haze pollution conditions. During the dust period, HGF reached a peak at 150nm and then declined as the particle size increased. The VSF for 50nm was the least, and increased with the increase of particle size from 150-300 nm, reaching as high as 0.74 at 300nm during the dust period. More discussion on aerosol hygroscopicity and volatility during the dust period will be given in the following section."*

Besides, we conducted a review of the existing studies on the hygroscopicity and

volatility of aerosols as reviewer's suggestion and modified the introduction as follow:

"*Hygroscopicity and volatility are critical physical properties of atmospheric aerosol particles. Hygroscopicity has a significant influence on atmospheric radiative balance and visibility by altering particle size distribution and optical properties. In addition, hygroscopicity indirectly affects the regional and global climate by influencing the lifetime and microphysical properties of clouds (Gunthe et al., 2009; Pöhlker et al., 2023; Rose et al., 2010). Moreover, hygroscopicity plays a key role in particle deposition by changing particle size in the human respiratory tract (Yu et al., 2025a). Volatility plays a crucial role in gas-particle partitioning, the formation and aging process of aerosols (Huffman et al., 2008; Xu et al., 2019). Considering the hygroscopicity and volatility of aerosols in the model is of great significance for reducing discrepancies between simulation results and observational data, and improving the accuracy of model outputs (Gao et al., 2024; Mcfiggans et al., 2006; Pringle et al., 2010; Rissler et al., 2010). Besides, determining the variation of particle size at selected dry diameters under different relative humidity and temperatures can also provide valuable indirect in-situ information regarding the chemical composition, mixing state, and coating properties of aerosols (Chen et al., 2022a; Liu et al., 2025; Massoli et al., 2010).*

*Hygroscopicity of aerosols have been investigated using several different instruments and techniques. The humidity tandem differential mobility analyzer (H-TDMA), which can provide information on the hygroscopic growth probability distribution of submicron aerosols, is widely used to measure aerosol hygroscopicity worldwide (Coe et al., 2007; Tang et al., 2019). In China, size-resolved aerosol hygroscopicity measurements have been carried out extensively in the North China Plain, Yangtze River Delta and Pearl River Delta. These researches focus on the seasonal variation of aerosol hygroscopicity (Zhang et al., 2023b; Wang et al., 2018b), the characteristics of hygroscopicity under different environment (Chen et al., 2022b; Wang et al., 2017), the impact of aerosol chemical composition and aging processes on aerosol hygroscopicity (Fan et al., 2020; Shi et al., 2022; Zhang et al., 2023a) and the evolution of hygroscopic behavior of atmospheric aerosols during heavy pollution episodes and new particle formation period (Wang et al., 2019b; Wu et al., 2017; Wu et al., 2016). Volatility tandem differential mobility analyzer (V-TDMA) is one of online instruments with high time resolution to measure the aerosol volatility. Multiple studies have been conducted in China using V-TDMA (Wang et al., 2017; Chen et al., 2022a; Wu et al., 2017) and inversed aerosol mixing state based on V-TDMA data (Chen et al.,*

*2020).*

*Aerosol hygroscopicity and volatility are correlated with chemical composition of the particles. Simultaneous measurements on aerosol hygroscopicity and volatility can provide new insight about the aging mechanisms of aerosols under different environment and the relationship of hygroscopicity and volatility. The VH-TDMA system was first proposed by Johnson et al. (2004), combining V-TDMA with H-TDMA to obtain both hygroscopicity and volatility simultaneously. Although aerosol hygroscopicity or volatility have been widely investigated worldwide, the simultaneous study of hygroscopicity and volatility in China is still limited (Cai et al., 2017; Kim et al., 2011; Wang et al., 2017; Yu et al., 2025a; Zhang et al., 2016). The results in the rural Pearl River Delta area reported that the photochemically-produced ultrafine particles to consist primarily of non-volatile and hygroscopic (NV-H) particles with a little volatile and non-hygroscopic (V-NH) particles and volatile and hygroscopic (V-H) particles(Kim et al., 2011). Zhang et al. (2016) found that certain fraction of hydrophobic particles is volatile in a rural site of the North China Plain. Wang et al. (2017) demonstrated that a higher number fraction of hydrophobic and volatile particles during the emission control period. The results observed by Yu et al. (2025b) showed that a positive correlation was identified between the number fraction of nearly hydrophobic and non-volatile particles during both the clean and the pollution periods.*

*Dust particles, suspended in the atmosphere, range from less than 0.1μm to over 100μm (Adebiyi et al., 2023). As one of the most important natural aerosols in the atmosphere, dust aerosols significantly affect atmospheric chemistry, human health, climate change, and biogeochemical cycles (Chen et al., 2021; Kurai et al., 2014). Heterogeneous reactions between mineral dust and trace gases can alter the chemical and physical properties of aerosols (Tang et al., 2017; Xu et al., 2020; Kok et al., 2023). Schladitz et al. (2011) demonstrated that the influence of dust particles was observed down to 300 nm during the Saharan Mineral Dust Experiment (SAMUM). Previous studies revealed that the changes of submicron aerosol effective density and optical properties during the dust period (Lu et al., 2024; Xia et al., 2019). Lu et al. (2024) found that the effective densities of 150, 250, 350, 450 nm under dusty conditions were higher than those during non-dusty periods, which reflected the dust influence on accumulation mode particles. Although the climatic and environmental effects of dust are considerable, limited studies focus on the dust effect on aerosol hygroscopicity and volatility simultaneously, especially on submicron aerosols (Kaaden et al., 2009; Kim and Park, 2012; Massling et al., 2007; Schladitz et al., 2011; Schladitz et al., 2009).*

*The North China Plain (NCP) is a hot spot of anthropogenic emissions, which can lead to severe air pollution. In recent years, the air quality in the NCP has significantly improved due to the strict control measures implemented by the government. However, the air pollution still happens due to unfavorable meteorological conditions, particularly in spring (Hu et al., 2021; Zhong et al., 2021). On the other hand, dust events often occur in the NCP in spring (Gui et al., 2023; Gui et al., 2022), which complicates the characteristics of aerosol properties in the NCP (Pan et al., 2009). Thus, it is necessary to enhance the comprehensive understanding of aerosol hygroscopicity and volatility in spring, particularly under different pollution conditions.*

*In this study, the hygroscopicity and volatility of aerosols were measured simultaneously using a Volatility-Hygroscopicity Tandem Differential Mobility Analyzer (VH-TDMA) in the spring of 2024. The characteristics of aerosol hygroscopicity and volatility were characterized, and the influence of air mass on aerosol hygroscopicity and volatility was discussed. Moreover, the relationships between hygroscopicity and volatility were explored. Besides, Aerosol hygroscopicity and volatility under different pollution environments were analyzed. Finally, the evolution of aerosol hygroscopicity, volatility and optical properties was investigated through a case study of a dust event. The study aimed to enhance understanding aerosol mixing state and evolution under different conditions and provide reliable observational constraints for reducing discrepancies between simulation results and observational data.*" (Line 41-130 in the revised manuscript)

Reference:

Adebiyi, A., Kok, J. F., Murray, B. J., Ryder, C. L., Stuut, J.-B. W., Kahn, R. A., Knippertz, P., Formenti, P., Mahowald, N. M., Pérez García-Pando, C., Klose, M., Ansmann, A., Samset, B. H., Ito, A., Balkanski, Y., Di Biagio, C., Romanias, M. N., Huang, Y., and Meng, J.: A review of coarse mineral dust in the Earth system, Aeolian Research, 60, 10.1016/j.aeolia.2022.100849, 2023.

Cai, M., Tan, H., Chan, C. K., Mochida, M., Hatakeyama, S., Kondo, Y., Schurman, M. I., Xu, H., Li, F., Shimada, K., Li, L., Deng, Y., Yai, H., Matsuki, A., Qin, Y., and Zhao, J.: Comparison of Aerosol Hygroscopcity, Volatility, and Chemical Composition between a Suburban Site in the Pearl River Delta Region and a Marine Site in Okinawa, Aerosol and Air Quality Research, 17, 3194-3208, 10.4209/aaqr.2017.01.0020, 2017.

Chen, L., Zhang, F., Collins, D., Ren, J., Liu, J., Jiang, S., and Li, Z.: Characterizing the volatility and mixing state of ambient fine particles in the summer and winter of urban Beijing, Atmospheric Chemistry and Physics, 22, 2293-2307, 10.5194/acp-22-2293-2022, 2022a.

Chen, L., Zhang, F., Yan, P., Wang, X., Sun, L., Li, Y., Zhang, X., Sun, Y., and Li, Z.:

The large proportion of black carbon (BC)-containing aerosols in the urban atmosphere, Environmental Pollution, 263, 10.1016/j.envpol.2020.114507, 2020.

Chen, L., Zhang, F., Zhang, D., Wang, X., Song, W., Liu, J., Ren, J., Jiang, S., Li, X., and Li, Z.: Measurement report: Hygroscopic growth of ambient fine particles measured at five sites in China, Atmospheric Chemistry and Physics, 22, 6773-6786, 10.5194/acp-22-6773-2022, 2022b.

Chen, S., Liu, J., Wang, X., Zhao, S., Chen, J., Qiang, M., Liu, B., Xu, Q., Xia, D., and Chen, F.: Holocene dust storm variations over northern China: transition from a natural forcing to an anthropogenic forcing, Science Bulletin, 66, 2516-2527, 10.1016/j.scib.2021.08.008, 2021.

Coe, H., Allan, J., Bower, K. N., Capes, G., Crosier, J., Haywood, J., Osborne, S., Minnikin, A., Murphy, J., Petzold, A., Reeves, C., and Williams, P.: Hygroscopic Properties of Sub-micrometer Atmospheric Aerosol Particles Measured with H-TDMA Instruments in Various Environments – A Review, Nucleation and Atmospheric Aerosols, Dordrecht, 716-720,

Fan, X., Liu, J., Zhang, F., Chen, L., Collins, D., Xu, W., Jin, X., Ren, J., Wang, Y., Wu, H., Li, S., Sun, Y., and Li, Z.: Contrasting size-resolved hygroscopicity of fine particles derived by HTDMA and HR-ToF-AMS measurements between summer and winter in Beijing: the impacts of aerosol aging and local emissions, Atmospheric Chemistry and Physics, 20, 915-929, 10.5194/acp-20-915-2020, 2020.

Feng, T., Wang, Y., Hu, W., Zhu, M., Song, W., Chen, W., Sang, Y., Fang, Z., Deng, W., Fang, H., Yu, X., Wu, C., Yuan, B., Huang, S., Shao, M., Huang, X., He, L., Lee, Y. R., Huey, L. G., Canonaco, F., Prevot, A. S. H., and Wang, X.: Impact of aging on the sources, volatility, and viscosity of organic aerosols in Chinese outflows, Atmospheric Chemistry and Physics, 23, 611-636, 10.5194/acp-23-611-2023, 2023.

Fu, Q., Zhuang, G., Li, J., Huang, K., Wang, Q., Zhang, R., Fu, J., Lu, T., Chen, M., Wang, Q., Chen, Y., Xu, C., and Hou, B.: Source, long‐range transport, and characteristics of a heavy dust pollution event in Shanghai, Journal of Geophysical Research: Atmospheres, 115, 10.1029/2009jd013208, 2010.

Gao, C. Y., Bauer, S. E., Tsigaridis, K., and Im, U.: Global Influence of Organic Aerosol Volatility on Aerosol Microphysical Processes: Composition and Number, Journal of Advances in Modeling Earth Systems, 16, 10.1029/2023ms004185, 2024.

Gui, K., Yao, W., Che, H., An, L., Zheng, Y., Li, L., Zhao, H., Zhang, L., Zhong, J., Wang, Y., and Zhang, X.: Record-breaking dust loading during two mega dust storm events over northern China in March 2021: aerosol optical and radiative properties and meteorological drivers, Atmospheric Chemistry and Physics, 22, 7905-7932, 10.5194/acp-22-7905-2022, 2022.

Gui, K., Che, H., Yao, W., Zheng, Y., Li, L., An, L., Wang, H., Wang, Y., Wang, Z., Ren, H.-L., Sun, J., Li, J., and Zhang, X.: Quantifying the contribution of local drivers to observed weakening of spring dust storm frequency over northern China (1982–2017), Science of The Total Environment, 894, 10.1016/j.scitotenv.2023.164923,

2023.

Gunthe, S. S., King, S. M., Rose, D., Chen, Q., Roldin, P., Farmer, D. K., Jimenez, J. L., Artaxo, P., Andreae, M. O., Martin, S. T., and Pöschl, U.: Cloud condensation nuclei in pristine tropical rainforest air of Amazonia: size-resolved measurements and modeling of atmospheric aerosol composition and CCN activity, Atmospheric Chemistry and Physics, 9, 7551-7575, DOI 10.5194/acp-9-7551-2009, 2009.

Hu, X., Sun, J., Xia, C., Shen, X., Zhang, Y., Zhang, X., and Zhang, S.: Simultaneous measurements of PM1 and PM10 aerosol scattering properties and their relationships in urban Beijing: A two-year observation, Science of The Total Environment, 770, 10.1016/j.scitotenv.2021.145215, 2021.

Huffman, J. A., Ziemann, P. J., Jayne, J. T., Worsnop, D. R., and Jimenez, J. L.: Development and Characterization of a Fast-Stepping/Scanning Thermodenuder for Chemically-Resolved Aerosol Volatility Measurements, Aerosol Science and Technology, 42, 395-407, 10.1080/02786820802104981, 2008.

Johnson, G. R., Ristovski, Z., and Morawska, L.: Method for measuring the hygroscopic behaviour of lower volatility fractions in an internally mixed aerosol, J Aerosol Sci, 35, 443-455, 10.1016/j.jaerosci.2003.10.008, 2004.

Kaaden, N., Massling, A., Schladitz, A., Müller, T., Kandler, K., Schütz, L., Weinzierl, B., Petzold, A., Tesche, M., Leinert, S., Deutscher, C., Ebert, M., Weinbruch, S., and Wiedensohler, A.: State of mixing, shape factor, number size distribution, and hygroscopic growth of the Saharan anthropogenic and mineral dust aerosol at Tinfou, Morocco, Tellus B: Chemical and Physical Meteorology, 61, 10.1111/j.1600-0889.2008.00388.x, 2009.

Kim, J.-S. and Park, K.: Atmospheric Aging of Asian Dust Particles During Long Range Transport, Aerosol Science and Technology, 46, 913-924, 10.1080/02786826.2012.680984, 2012.

Kim, J.-S., Kim, Y. J., and Park, K.: Measurements of hygroscopicity and volatility of atmospheric ultrafine particles in the rural Pearl River Delta area of China, Atmospheric Environment, 45, 4661-4670, 10.1016/j.atmosenv.2011.05.054, 2011.

Kurai, J., Watanabe, M., Tomita, K., Yamasaki, H. S. A., and Shimizu, E.: Influence of Asian Dust Particles on Immune Adjuvant Effects and Airway Inflammation in Asthma Model Mice, PLoS ONE, 9, 10.1371/journal.pone.0111831, 2014.

Liu, J., Zhang, F., Xu, W., Chen, L., Ren, J., Jiang, S., Sun, Y., and Li, Z.: A Large Impact of Cooking Organic Aerosol (COA) on Particle Hygroscopicity and CCN Activity in Urban Atmosphere, Journal of Geophysical Research: Atmospheres, 126, 10.1029/2020jd033628, 2021.

Liu, J., Zhang, F., Ren, J., Chen, L., Zhang, A., Wang, Z., Zou, S., Xu, H., and Yue, X.: The evolution of aerosol mixing state derived from a field campaign in Beijing: implications for particle aging timescales in urban atmospheres, Atmospheric Chemistry and Physics, 25, 5075-5086, 10.5194/acp-25-5075-2025, 2025.

Liu, T., Duan, F., Ma, Y., Ma, T., Zhang, Q., Xu, Y., Li, F., Huang, T., Kimoto, T., Zhang, Q., and He, K.: Classification and sources of extremely severe sandstorms mixed with haze pollution in Beijing, Environmental Pollution, 322,

10.1016/j.envpol.2023.121154, 2023.

Lu, J., Shen, X., Ma, Q., Yu, A., Hu, X., Zhang, Y., Liu, Q., Liu, S., Che, H., Zhang, X., and Sun, J.: Size-resolved effective density of ambient aerosols measured by an AAC–SMPS tandem system in Beijing, Atmospheric Environment, 318, 10.1016/j.atmosenv.2023.120226, 2024.

Massling, A., Leinert, S., Wiedensohler, A., and Covert, D.: Hygroscopic growth of sub-micrometer and one-micrometer aerosol particles measured during ACE-Asia, Atmospheric Chemistry and Physics, 7, 3249-3259, DOI 10.5194/acp-7-3249-2007, 2007.

Massoli, P., Lambe, A. T., Ahern, A. T., Williams, L. R., Ehn, M., Mikkilä, J., Canagaratna, M. R., Brune, W. H., Onasch, T. B., Jayne, J. T., Petäjä, T., Kulmala, M., Laaksonen, A., Kolb, C. E., Davidovits, P., and Worsnop, D. R.: Relationship between aerosol oxidation level and hygroscopic properties of laboratory generated secondary organic aerosol (SOA) particles, Geophysical Research Letters, 37, 10.1029/2010gl045258, 2010.

McFiggans, G., Artaxo, P., Baltensperger, U., Coe, H., Facchini, M. C., Feingold, G., Fuzzi, S., Gysel, M., Laaksonen, A., Lohmann, U., Mentel, T. F., Murphy, D. M., O'Dowd, C. D., Snider, J. R., and Weingartner, E.: The effect of physical and chemical aerosol properties on warm cloud droplet activation, Atmospheric Chemistry and Physics, 6, 2593-2649, DOI 10.5194/acp-6-2593-2006, 2006.

Pan, X. L., Yan, P., Tang, J., Ma, J. Z., Wang, Z. F., Gbaguidi, A., and Sun, Y. L.: Observational study of influence of aerosol hygroscopic growth on scattering coefficient over rural area near Beijing mega-city, Atmospheric Chemistry and Physics, 9, 7519-7530, DOI 10.5194/acp-9-7519-2009, 2009.

Pöhlker, M. L., Pöhlker, C., Quaas, J., Mülmenstädt, J., Pozzer, A., Andreae, M. O., Artaxo, P., Block, K., Coe, H., Ervens, B., Gallimore, P., Gaston, C. J., Gunthe, S. S., Henning, S., Herrmann, H., Krüger, O. O., McFiggans, G., Poulain, L., Raj, S. S., Reyes-Villegas, E., Royer, H. M., Walter, D., Wang, Y., and Pöschl, U.: Global organic and inorganic aerosol hygroscopicity and its effect on radiative forcing, Nature Communications, 14, 10.1038/s41467-023-41695-8, 2023.

Pringle, K. J., Tost, H., Pozzer, A., Pöschl, U., and Lelieveld, J.: Global distribution of the effective aerosol hygroscopicity parameter for CCN activation, Atmospheric Chemistry and Physics, 10, 5241-5255, 10.5194/acp-10-5241-2010, 2010.

Rissler, J., Svenningsson, B., Fors, E. O., Bilde, M., and Swietlicki, E.: An evaluation and comparison of cloud condensation nucleus activity models: Predicting particle critical saturation from growth at subsaturation, Journal of Geophysical Research: Atmospheres, 115, 10.1029/2010jd014391, 2010.

Rose, D., Nowak, A., Achtert, P., Wiedensohler, A., Hu, M., Shao, M., Zhang, Y., Andreae, M. O., and Pöschl, U.: Cloud condensation nuclei in polluted air and biomass burning smoke near the mega-city Guangzhou, China - Part 1: Size-resolved measurements and implications for the modeling of aerosol particle hygroscopicity and CCN activity, Atmospheric Chemistry and Physics, 10, 3365-3383, DOI 10.5194/acp-10-3365-2010, 2010.

Schladitz, A., Müller, T., Nowak, A., Kandler, K., Lieke, K., Massling, A., and

Wiedensohler, A.: In situ aerosol characterization at Cape Verde: Part 1: Particle number size distributions, hygroscopic growth and state of mixing of the marine and Saharan dust aerosol, Tellus B: Chemical and Physical Meteorology, 63, 10.1111/j.1600-0889.2011.00569.x, 2011.

Schladitz, A., Müller, T., Kaaden, N., Massling, A., Kandler, K., Ebert, M., Weinbruch, S., Deutscher, C., and Wiedensohler, A.: In situ measurements of optical properties at Tinfou (Morocco) during the Saharan Mineral Dust Experiment SAMUM 2006, Tellus B: Chemical and Physical Meteorology, 61, 10.1111/j.1600-0889.2008.00397.x, 2009.

Shi, J., Hong, J., Ma, N., Luo, Q., He, Y., Xu, H., Tan, H., Wang, Q., Tao, J., Zhou, Y., Han, S., Peng, L., Xie, L., Zhou, G., Xu, W., Sun, Y., Cheng, Y., and Su, H.: Measurement report: On the difference in aerosol hygroscopicity between high and low relative humidity conditions in the North China Plain, Atmospheric Chemistry and Physics, 22, 4599-4613, 10.5194/acp-22-4599-2022, 2022.

Song, X., Wang, Y., Huang, X., Wang, Y., Li, Z., Zhu, B., Ren, R., An, J., Yan, J., Zhang, R., Shang, Y., and Zhan, P.: The Impacts of Dust Storms With Different Transport Pathways on Aerosol Chemical Compositions and Optical Hygroscopicity of Fine Particles in the Yangtze River Delta, Journal of Geophysical Research: Atmospheres, 128, 10.1029/2023jd039679, 2023.

Sun, Y., Du, W., Fu, P., Wang, Q., Li, J., Ge, X., Zhang, Q., Zhu, C., Ren, L., Xu, W., Zhao, J., Han, T., Worsnop, D. R., and Wang, Z.: Primary and secondary aerosols in Beijing in winter: sources, variations and processes, Atmospheric Chemistry and Physics, 16, 8309-8329, 10.5194/acp-16-8309-2016, 2016.

Tang, M., Chan, C. K., Li, Y. J., Su, H., Ma, Q., Wu, Z., Zhang, G., Wang, Z., Ge, M., Hu, M., He, H., and Wang, X.: A review of experimental techniques for aerosol hygroscopicity studies, Atmospheric Chemistry and Physics, 19, 12631-12686, 10.5194/acp-19-12631-2019, 2019.

Tiitta, P., Miettinen, P., Vaattovaara, P., Joutsensaari, J., Petäjä, T., Virtanen, A., Raatikainen, T., Aalto, P., Portin, H., and Romakkaniemi, S.: Roadside aerosol study using hygroscopic, organic and volatility TDMAs: Characterization and mixing state, Atmospheric Environment, 44, 976-986, 10.1016/j.atmosenv.2009.06.021, 2010.

Tong, D. Q., Wang, J. X. L., Gill, T. E., Lei, H., and Wang, B.: Intensified dust storm activity and Valley fever infection in the southwestern United States, Geophysical Research Letters, 44, 4304-4312, 10.1002/2017gl073524, 2017.

Wang, X., Zhang, R., and Yu, W.: The Effects of PM2.5 Concentrations and Relative Humidity on Atmospheric Visibility in Beijing, Journal of Geophysical Research: Atmospheres, 124, 2235-2259, 10.1029/2018jd029269, 2019a.

Wang, X., Shen, X. J., Sun, J. Y., Zhang, X. Y., Wang, Y. Q., Zhang, Y. M., Wang, P., Xia, C., Qi, X. F., and Zhong, J. T.: Size-resolved hygroscopic behavior of atmospheric aerosols during heavy aerosol pollution episodes in Beijing in December 2016, Atmospheric Environment, 194, 188-197, 10.1016/j.atmosenv.2018.09.041, 2018a.

Wang, Y., Wu, Z., Ma, N., Wu, Y., Zeng, L., Zhao, C., and Wiedensohler, A.: Statistical

analysis and parameterization of the hygroscopic growth of the sub-micrometer urban background aerosol in Beijing, Atmospheric Environment, 175, 184-191, 10.1016/j.atmosenv.2017.12.003, 2018b.

Wang, Y., Zhang, F., Li, Z., Tan, H., Xu, H., Ren, J., Zhao, J., Du, W., and Sun, Y.: Enhanced hydrophobicity and volatility of submicron aerosols under severe emission control conditions in Beijing, Atmospheric Chemistry and Physics, 17, 5239-5251, 10.5194/acp-17-5239-2017, 2017.

Wang, Y., Li, Z., Zhang, R., Jin, X., Xu, W., Fan, X., Wu, H., Zhang, F., Sun, Y., Wang, Q., Cribb, M., and Hu, D.: Distinct Ultrafine‐ and Accumulation‐Mode Particle Properties in Clean and Polluted Urban Environments, Geophysical Research Letters, 46, 10918-10925, 10.1029/2019gl084047, 2019b.

Wang, Y. Q., Zhang, X. Y., Sun, J. Y., Zhang, X. C., Che, H. Z., and Li, Y.: Spatial and temporal variations of the concentrations of PM10, PM2.5 and PM1 in China, Atmospheric Chemistry and Physics, 15, 13585-13598, 10.5194/acp-15-13585-2015, 2015.

Wu, Z. J., Zheng, J., Shang, D. J., Du, Z. F., Wu, Y. S., Zeng, L. M., Wiedensohler, A., and Hu, M.: Particle hygroscopicity and its link to chemical composition in the urban atmosphere of Beijing, China, during summertime, Atmospheric Chemistry and Physics, 16, 1123-1138, 10.5194/acp-16-1123-2016, 2016.

Wu, Z. J., Ma, N., Größ, J., Kecorius, S., Lu, K. D., Shang, D. J., Wang, Y., Wu, Y. S., Zeng, L. M., Hu, M., Wiedensohler, A., and Zhang, Y. H.: Thermodynamic properties of nanoparticles during new particle formation events in the atmosphere of North China Plain, Atmospheric Research, 188, 55-63, 10.1016/j.atmosres.2017.01.007, 2017.

Xia, C., Sun, J., Qi, X., Shen, X., Zhong, J., Zhang, X., Wang, Y., Zhang, Y., and Hu, X.: Observational study of aerosol hygroscopic growth on scattering coefficient in Beijing: A case study in March of 2018, Science of The Total Environment, 685, 239-247, 10.1016/j.scitotenv.2019.05.283, 2019.

Xu, W., Xie, C., Karnezi, E., Zhang, Q., Wang, J., Pandis, S. N., Ge, X., Zhang, J., An, J., Wang, Q., Zhao, J., Du, W., Qiu, Y., Zhou, W., He, Y., Li, Y., Li, J., Fu, P., Wang, Z., Worsnop, D. R., and Sun, Y.: Summertime aerosol volatility measurements in Beijing, China, Atmospheric Chemistry and Physics, 19, 10205-10216, 10.5194/acp-19-10205-2019, 2019.

Yu, A., Lu, J., Shen, X., Hu, X., Zhang, Y., Liu, Q., Tong, H., Liang, L., Liu, L., Ma, Q., Han, L., Che, H., Zhang, X., and Sun, J.: Determination of the deposition of urban submicron aerosols in the human respiratory tract considering hygroscopic growth, Atmospheric Environment, 356, 10.1016/j.atmosenv.2025.121289, 2025a.

Yu, A., Shen, X., Ma, Q., Lu, J., Hu, X., Zhang, Y., Liu, Q., Liang, L., Liu, L., Liu, S., Tong, H., Che, H., Zhang, X., and Sun, J.: Size-resolved hygroscopicity and volatility properties of ambient urban aerosol particles measured by a volatility hygroscopicity tandem differential mobility analyzer system in Beijing, Atmospheric Chemistry and Physics, 25, 3389-3412, 10.5194/acp-25-3389-2025, 2025b.

Zhang, S., Shen, X., Sun, J., Zhang, Y., Zhang, X., Xia, C., Hu, X., Zhong, J., Wang, J., and Liu, S.: Atmospheric Particle Hygroscopicity and the Influence by Oxidation State of Organic Aerosols in Urban Beijing, Journal of Environmental Sciences, 124, 544-556, 10.1016/j.jes.2021.11.019, 2023a.

Zhang, S., Shen, X., Sun, J., Che, H., Zhang, Y., Liu, Q., Xia, C., Hu, X., Zhong, J., Wang, J., Liu, S., Lu, J., Yu, A., and Zhang, X.: Seasonal variation of particle hygroscopicity and its impact on cloud-condensation nucleus activation in the Beijing urban area, Atmospheric Environment, 302, 10.1016/j.atmosenv.2023.119728, 2023b.

Zhang, S. L., Ma, N., Kecorius, S., Wang, P. C., Hu, M., Wang, Z. B., Größ, J., Wu, Z. J., and Wiedensohler, A.: Mixing state of atmospheric particles over the North China Plain, Atmospheric Environment, 125, 152-164, 10.1016/j.atmosenv.2015.10.053, 2016.

Zhong, J., Zhang, X., Wang, Y., Sun, J., Shen, X., Xia, C., and Zhang, W.: Attribution of the worse aerosol pollution in March 2018 in Beijing to meteorological variability, Atmospheric Research, 250, 10.1016/j.atmosres.2020.105294, 2021.

---

## Author Comment (AC2)

**Response to Reviewer #3**

The manuscript describes an intense measurement period in spring in the North China Plateau (NCP) focusing on sub-micrometre hygroscopicity and volatility measurements. They performed Hysplit back-trajectory analysis and in combination with PM10 and PM2.5 values, determined a special period with most likely dust aerosols from the Gobi desert. The publication is meant as a measurement report rather than a full research article. Generally, characterizing aerosols' water uptake ability and volatility is very important to better constrain model parametrizations. To my mind the article lacks some more discussion regarding what has previously been measured in the NCP and puts very much focus on a single dust event that occurred during the measurement period. I also think, it would be helpful to put the dust period in perspective to other dust events when hygroscopicity and volatility were measured, even outside of the NCP. Furthermore, the article needs a major revision regarding English spelling and structure.

**Reply:** Thanks for reviewer's constructive suggestions and comments. To strengthen the contextualization of our work within existing literature, we modified Introduction section to synthesize previous measurements of aerosol hygroscopicity and volatility in the China and clarified the importance of this researches. Please see more details in Line 40-130, also shown below:

*"Hygroscopicity and volatility are critical physical properties of atmospheric aerosol particles. Hygroscopicity has a significant influence on atmospheric radiative balance and visibility by altering particle size distribution and optical properties. In addition, hygroscopicity indirectly affects the regional and global climate by influencing the lifetime and microphysical properties of clouds (Gunthe et al., 2009; Pöhlker et al., 2023; Rose et al., 2010). Moreover, hygroscopicity plays a key role in particle deposition by changing particle size in the human respiratory tract (Yu et al., 2025a). Volatility plays a crucial role in gas-particle partitioning, the formation and aging process of aerosols (Huffman et al., 2008; Xu et al., 2019). Considering the hygroscopicity and volatility of aerosols in the model is of great significance for reducing discrepancies between simulation results and observational data, and improving the accuracy of model outputs (Gao et al., 2024; Mcfiggans et al., 2006; Pringle et al., 2010; Rissler et al., 2010). Besides, determining the variation of particle size at selected dry diameters under different relative humidity and temperatures can*

*also provide valuable indirect in-situ information regarding the chemical composition, mixing state, and coating properties of aerosols (Chen et al., 2022a; Liu et al., 2025; Massoli et al., 2010).*

*Hygroscopicity of aerosols have been investigated using several different instruments and techniques. The humidity tandem differential mobility analyzer (H-TDMA), which can provide information on the hygroscopic growth probability distribution of submicron aerosols, is widely used to measure aerosol hygroscopicity worldwide (Coe et al., 2007; Tang et al., 2019). In China, size-resolved aerosol hygroscopicity measurements have been carried out extensively in the North China Plain, Yangtze River Delta and Pearl River Delta. These researches focus on the seasonal variation of aerosol hygroscopicity (Zhang et al., 2023b; Wang et al., 2018), the characteristics of hygroscopicity under different environment (Chen et al., 2022b; Wang et al., 2017), the impact of aerosol chemical composition and aging processes on aerosol hygroscopicity (Fan et al., 2020; Shi et al., 2022; Zhang et al., 2023a) and the evolution of hygroscopic behavior of atmospheric aerosols during heavy pollution episodes and new particle formation period (Wang et al., 2019; Wu et al., 2017; Wu et al., 2016). Volatility tandem differential mobility analyzer (V-TDMA) is one of online instruments with high time resolution to measure the aerosol volatility. Multiple studies have been conducted in China using V-TDMA (Wang et al., 2017; Chen et al., 2022a; Wu et al., 2017) and inversed aerosol mixing state based on V-TDMA data (Chen et al., 2020).*

*Aerosol hygroscopicity and volatility are correlated with chemical composition of the particles. Simultaneous measurements on aerosol hygroscopicity and volatility can provide new insight about the aging mechanisms of aerosols under different environment and the relationship of hygroscopicity and volatility. The VH-TDMA system was first proposed by Johnson et al. (2004), combining V-TDMA with H-TDMA to obtain both hygroscopicity and volatility simultaneously. Although aerosol hygroscopicity or volatility have been widely investigated worldwide, the simultaneous study of hygroscopicity and volatility in China is still limited (Cai et al., 2017; Kim et al., 2011; Wang et al., 2017; Yu et al., 2025a; Zhang et al., 2016). The results in the rural Pearl River Delta area reported that the photochemically-produced ultrafine particles to consist primarily of non-volatile and hygroscopic (NV-H) particles with a little volatile and non-hygroscopic (V-NH) particles and volatile and hygroscopic (V-H) particles(Kim et al., 2011). Zhang et al. (2016) found that certain fraction of hydrophobic particles is volatile in a rural site of the North China Plain. Wang et al.*

*(2017) demonstrated that a higher number fraction of hydrophobic and volatile particles during the emission control period. The results observed by Yu et al. (2025b) showed that a positive correlation was identified between the number fraction of nearly hydrophobic and non-volatile particles during both the clean and the pollution periods.*

*Dust particles, suspended in the atmosphere, range from less than 0.1μm to over 100μm (Adebiyi et al., 2023). As one of the most important natural aerosols in the atmosphere, dust aerosols significantly affect atmospheric chemistry, human health, climate change, and biogeochemical cycles (Chen et al., 2021; Kurai et al., 2014). Heterogeneous reactions between mineral dust and trace gases can alter the chemical and physical properties of aerosols (Tang et al., 2017; Xu et al., 2020; Kok et al., 2023). Schladitz et al. (2011) demonstrated that the influence of dust particles was observed down to 300 nm during the Saharan Mineral Dust Experiment (SAMUM). Previous studies revealed that the changes of submicron aerosol effective density and optical properties during the dust period (Lu et al., 2024; Xia et al., 2019). Lu et al. (2024) found that the effective densities of 150, 250, 350, 450 nm under dusty conditions were higher than those during non-dusty periods, which reflected the dust influence on accumulation mode particles. Although the climatic and environmental effects of dust are considerable, limited studies focus on the dust effect on aerosol hygroscopicity and volatility simultaneously, especially on submicron aerosols (Kaaden et al., 2009; Kim and Park, 2012; Massling et al., 2007; Schladitz et al., 2011; Schladitz et al., 2009).*

*The North China Plain (NCP) is a hot spot of anthropogenic emissions, which can lead to severe air pollution. In recent years, the air quality in the NCP has significantly improved due to the strict control measures implemented by the government. However, the air pollution still happens due to unfavorable meteorological conditions, particularly in spring (Hu et al., 2021; Zhong et al., 2021). On the other hand, dust events often occur in the NCP in spring (Gui et al., 2023; Gui et al., 2022), which complicates the characteristics of aerosol properties in the NCP (Pan et al., 2009). Thus, it is necessary to enhance the comprehensive understanding of aerosol hygroscopicity and volatility in spring, particularly under different pollution conditions.*

*In this study, the hygroscopicity and volatility of aerosols were measured simultaneously using a Volatility-Hygroscopicity Tandem Differential Mobility Analyzer (VH-TDMA) in the spring of 2024. The characteristics of aerosol hygroscopicity and volatility were characterized, and the influence of air mass on aerosol hygroscopicity and volatility was discussed. Moreover, the relationships between hygroscopicity and volatility were explored. Besides, Aerosol hygroscopicity*

*and volatility under different pollution environments were analyzed. Finally, the evolution of aerosol hygroscopicity, volatility and optical properties was investigated through a case study of a dust event. The study aimed to enhance understanding aerosol mixing state and evolution under different conditions and provide reliable observational constraints for reducing discrepancies between simulation results and observational data."*

As reviewer's suggestion, we added the comparison between our findings during the dust period with previous studies as follow:

*"The averages of SAE for $PM_1$ and $PM_{10}$ decreased sharply from 1.69 and 1.27 before dust period to 0.02 and -0.16 during the dust period, suggested a significant shift in aerosol size distribution toward coarse particle dominance (Hu et al., 2021). The persistently low SAE for $PM_1$ and $PM_{10}$ values throughout the dust episode suggested the larger particles were dominated. . During the dust period, SAE was comparable to previous results during the dust periods in Beijing and Nanjing (Song et al., 2023; Xia et al., 2019). The mean f(80%) for $PM_1$ and $PM_{10}$ decreased from 1.62 and 1.60 before the dust to 1.03 and 1.02 during the dust period, suggested that the submicron and super micron aerosols are almost hydrophobic during the dust period. The f(80%) during the dust period was similar to the value that observed in Beijing dust period (Xia et al., 2019). Although the scattering coefficient for $PM_1$ and $PM_{10}$ still remained a low level after the dust period, SAE and f(80%) gradually rose, implying that the fine particles from anthropogenic sources gradually became dominant."* (Line 484-497 in the revised manuscript)

*"During the dust period, the proportion of more hygroscopic mode particles decreased rapidly. When the dust arrived, the strong north wind not only brought dust particles but also swept pre-existing fine particulate matter in the atmosphere. The most significant decline in the MH mode number fraction occurred at 200 nm and 300 nm, with minimum values of 0.20 and 0.32, respectively. The mean MH mode number fractions for 200 and 300nm were 0.54 and 0.33, which were much lower than those before the dust period. Previous research also found that the number fractions of MH mode particles for 250 and 350 nm were less than 0.5 during the dust event observed in Tinfou, Morocco (Kaaden et al., 2009). Massling et al. (2007) found that the number fraction of nearly hydrophobic particle from dust particles was approximately 64%, which was consistent with our observation. In terms of volatility, the NV mode of 200*

*and 300 nm became more prominent during dust period. The mean values of the number fraction of VV mode for 200 and 300 nm particles decreased from 0.93 and 0.94 before the dust period to 0.73 and 0.47 during the dust period, respectively. The minimum NFvv at 300 nm was only 0.18, indicating that approximately 82% of particles with diameter of 300 nm were non-volatile. High number fraction of non-volatile components and nearly hydrophobic particles at 200 and 300 nm during the dust period suggested dust particles can be as small as 200 nm in diameter (Kaaden et al., 2009). The mean HGF for 300 nm particles during the dust period was 1.20, which was close to the observed results of Massling et al. (2007) during the dust period. Besides, low hygroscopicity of aerosol during dust storm was also observed by Shen et al. (2023). VSF increased evidently to the maximum value of 0.89 at 300 nm during the dust period."* (Line 511-532 in the revised manuscript)

Besides, we have thoroughly revised the manuscript with attention to grammatical accuracy and reorganized the manuscript thoroughly.

My major concerns are the following:

• The motivation of the study is lacking. The authors' refer to previous publications presenting the aerosols' properties, including hygroscopicity and volatility, in the same area and state that those previous studies related their results to chemical composition. As a major new thing, they state the single dust event. I cannot clearly see how this dataset differs from the previous ones. The authors state that these are the first hygroscopicity and volatility measurements in NCP during a dust event. After a quick literature research, I found several papers discussing dust events in NCP, which are though only partly mentioned in the paper under review. I think it would be very interesting to relate the new data to previous events and for example compare PM values and meteorological conditions and potentially relate the chemical composition measured previously with the hygroscopic growth found in this study. On the other hand, I believe that the dataset also offers other interesting discussion that could be deepened.

**Reply:** Thanks for reviewer's suggestions. we have revised the introduction thoroughly and added a paragraph in the revised manuscript to explicitly clarify the motivation of study. Aerosol properties are rather complex in the Beijing of Spring because of the mixture of different sources from anthropogenic emissions and dust while previous studies have characterized aerosol hygroscopicity and volatility, limited

researches focused on aerosol hygroscopicity and volatility under different conditions simultaneously. Besides, the understanding of dust effect on submicron aerosol remains inadequately understood. We have restructured the manuscript and enhanced the comparison and discussion with the results of previous studies. Moreover, We added the summary of meteorological parameters under different pollution conditions in the supplement (Table. S1) and linked the PM ratio and key meteorological parameters (relative humidity, wind speed, wind direction) from the current study with previous studies as follow:

*"There were six haze pollution events and one dust pollution event during the study period. The haze pollution events were characterized by high $PM_{2.5}$ mass loadings and high $PM_{2.5}$ and $PM_{10}$ ratio (0.46-0.72), which was similar to the results observed in Beijing (Liang et al., 2022). These six haze pollution events were strongly associated with the stagnant meteorological conditions as indicated by the prevailing southerly winds, high RH (>60%), and low wind speed (<1.3m/s) (Table S1). Previous studies in Beijing have shown that aerosol pollution in Beijing possibly contributed by pollutants transported from the south of Beijing (Wang et al., 2013; Yin et al., 2025). Furthermore, low wind speeds favored pollutant accumulation and high relative humidity would further enhance aerosol hygroscopic growth and accelerate liquid-phase and heterogeneous reactions (Zhong et al., 2018). In contrast, during the dust period, the average $PM_{2.5}$ mass concentration was only about half of that observed during haze episodes, whereas the $PM_{10}$ mass loading was approximately twice as high. Besides, low $PM_{2.5}$ and $PM_{10}$ ratio (0.16) was observed during dust period, which was comparable with that observed in Beijing dust period (Lu et al., 2024; Xia et al., 2019; Liang et al., 2022). The dust event was also associated with prevailing northerly winds, low RH (17%), and higher wind speeds (3.5 m/s) (Table S1), consistent with the typical meteorological conditions observed during dust period in previous studies (Lu et al., 2024)."* (Line 254-272 in the revised manuscript)

Table S1. Summary of meteorological parameters under different pollution conditions

| | T(℃) | RH (%) | Wind speed (m/s) | $PM_{2.5}$ ($\mu g/m^3$) | $PM_{10}$ ($\mu g/m^3$) | $PM_{2.5}/PM_{10}$ |
|---|---|---|---|---|---|---|
| EP1 | 14.2±5.2 | 63±19 | 1.03±1 | 92.2±16.5 | 166.2±37.6 | 0.56±0.04 |
| EP2 | 17.5±4.1 | 66±15 | 1.26±0.7 | 128.8±38.4 | 181.4±60.3 | 0.72±0.06 |
| DS | 18.7±2.1 | 17±4 | 3.5±0.6 | 49.3±19.9 | 297.1±107 | 0.16±0.02 |
| EP3 | 15.4±5.2 | 62±20 | 1.15±1.01 | 86.5±6.6 | 158.9±8.8 | 0.54±0.03 |

| | | | | | | |
|---|---|---|---|---|---|---|
| EP4 | 19.5±2.9 | 67±20 | 1.16±0.39 | 81.9±5.5 | 134.5±6.3 | 0.61±0.02 |
| EP5 | 16.1±3.4 | 76±15 | 1.06±0.55 | 81.6±5.1 | 147.6±8.8 | 0.55±0.04 |
| EP6 | 19.6±3.2 | 80±12 | 0.85±0.57 | 86.3±12.2 | 189.9±28.9 | 0.46±0.04 |

Besides, we compared our observational results with previous studies and analyzed the differences in measurements between the two regions based on chemical composition discrepancy in different region. Please see more details in Line 286-295 and 318-327, also shown below:

*"During the whole period, the number fraction of more hygroscopic mode particles (NF$_{MH}$) for 50-300 nm particles was usually larger than 0.50, especially at large size, suggested that more hygroscopic mode particles were dominated. While the results observed in Gucheng, an rural site in the North China Plain, showed that the number fraction of NH mode became more prominent with increasing particle size (Shi et al., 2022). The discrepancy in the size dependency of NF$_{MH}$ in Beijing and Gucheng is likely due to differences in chemical composition. Organics constituted a major fraction of PM$_1$ at Gucheng (Shi et al., 2022) , whereas secondary inorganic aerosols with high hygroscopicity made the largest contribution to PM$_1$ in Beijing (Lei et al., 2021)."*

*"During the sampling period, the mean number fraction of very volatile mode particles (NF$_{VV}$) for 50, 80, 110, 150, 200 and 300 nm particles was 0.96, 0.95, 0.94, 0.93, 0.91, 0.87, respectively, which were much higher than those measured at Xianghe, a rural site in the North China Plain (Zhang et al., 2016). In other words, the mean number fraction of non-volatile mode particles in Beijing was less than10%, which was lower than the observation results in Guangzhou (Cai et al., 2017) and Shanghai (Jiang et al., 2018). The high NF$_{VV}$ indicated that the majority of particles in Beijing were highly volatile, likely attributable to a lower proportion of black carbon and a higher proportion of secondary inorganic aerosols, especially nitrate (Lei et al., 2021; Sun et al., 2022)."*

I believe that the title is misleading as dust is only a very short period during the whole measurement period.

**Reply:** Thanks for reviewer's suggestion. We have changed the title of this manuscript to "Measurement report: Characteristics of hygroscopicity and volatility of submicron aerosols under different pollution environment in Spring".

• When discussing the differences in measured properties as a function of particle diameter, the authors make a differentiation between organic particles and "secondary particles" that are however, not specified in their composition. From the citation it looks to me like they are referring to "secondary organic aerosols". If this is so, I cannot understand the explanation for the differences between Aitken and accumulation mode particles. The observed differences with size are very interesting but are lacking further investigation.

**Reply:** We are sorry that we did not express this clearly. We have revised the manuscript as follow:

*"During the whole period, the number fraction of more hygroscopic mode particles ($NF_{MH}$) for 50-300 nm particles was usually larger than 0.50, especially at large size, suggested that more hygroscopic mode particles were dominated. While the results observed in Gucheng, an rural site in the North China Plain, showed that the number fraction of NH mode became more prominent with increasing particle size (Shi et al., 2022). The discrepancy in the size dependency of $NF_{MH}$ in Beijing and Gucheng is likely due to differences in chemical composition. Organics constituted a major fraction of $PM_1$ at Gucheng (Shi et al., 2022) , whereas secondary inorganic aerosols with high hygroscopicity made the largest contribution to $PM_1$ in Beijing (Lei et al., 2021)."* (Line 286-295 in the revised manuscript)

*"This distinct relationship between hygroscopicity and volatility of accumulation mode particles, compared to Aitken mode particles, is likely associated with differences in aerosol chemical composition. Previous studies in Beijing revealed that the particles below 100nm are mainly organics (Li et al., 2023), while accumulation mode particles are dominated by secondary inorganic aerosols characterized by high hygroscopicity and high volatility (Xu et al., 2015)."* (Line 387-393 in the revised manuscript).

Overall I believe that this manuscript should only be considered for publication in ACP after major revisions.

**Reply:** Thanks for reviewer's suggestion. We have reorganized the manuscript thoroughly and have done our best to enhance manuscript's quality.

**Reference:**

Adebiyi, A., Kok, J. F., Murray, B. J., Ryder, C. L., Stuut, J.-B. W., Kahn, R. A., Knippertz, P., Formenti, P., Mahowald, N. M., Pérez García-Pando, C., Klose, M., Ansmann, A., Samset, B. H., Ito, A., Balkanski, Y., Di Biagio, C., Romanias, M. N., Huang, Y., and Meng, J.: A review of coarse mineral dust in the Earth system, Aeolian Research, 60, 10.1016/j.aeolia.2022.100849, 2023.

Cai, M., Tan, H., Chan, C. K., Mochida, M., Hatakeyama, S., Kondo, Y., Schurman, M. I., Xu, H., Li, F., Shimada, K., Li, L., Deng, Y., Yai, H., Matsuki, A., Qin, Y., and Zhao, J.: Comparison of Aerosol Hygroscopcity, Volatility, and Chemical Composition between a Suburban Site in the Pearl River Delta Region and a Marine Site in Okinawa, Aerosol and Air Quality Research, 17, 3194-3208, 10.4209/aaqr.2017.01.0020, 2017.

Chen, L., Zhang, F., Collins, D., Ren, J., Liu, J., Jiang, S., and Li, Z.: Characterizing the volatility and mixing state of ambient fine particles in the summer and winter of urban Beijing, Atmospheric Chemistry and Physics, 22, 2293-2307, 10.5194/acp-22-2293-2022, 2022a.

Chen, L., Zhang, F., Yan, P., Wang, X., Sun, L., Li, Y., Zhang, X., Sun, Y., and Li, Z.: The large proportion of black carbon (BC)-containing aerosols in the urban atmosphere, Environmental Pollution, 263, 10.1016/j.envpol.2020.114507, 2020.

Chen, L., Zhang, F., Zhang, D., Wang, X., Song, W., Liu, J., Ren, J., Jiang, S., Li, X., and Li, Z.: Measurement report: Hygroscopic growth of ambient fine particles measured at five sites in China, Atmospheric Chemistry and Physics, 22, 6773-6786, 10.5194/acp-22-6773-2022, 2022b.

Chen, S., Liu, J., Wang, X., Zhao, S., Chen, J., Qiang, M., Liu, B., Xu, Q., Xia, D., and Chen, F.: Holocene dust storm variations over northern China: transition from a natural forcing to an anthropogenic forcing, Science Bulletin, 66, 2516-2527, 10.1016/j.scib.2021.08.008, 2021.

Coe, H., Allan, J., Bower, K. N., Capes, G., Crosier, J., Haywood, J., Osborne, S., Minnikin, A., Murphy, J., Petzold, A., Reeves, C., and Williams, P.: Hygroscopic Properties of Sub-micrometer Atmospheric Aerosol Particles Measured with H-TDMA Instruments in Various Environments – A Review, Nucleation and Atmospheric Aerosols, Dordrecht, 716-720,

Fan, X., Liu, J., Zhang, F., Chen, L., Collins, D., Xu, W., Jin, X., Ren, J., Wang, Y., Wu, H., Li, S., Sun, Y., and Li, Z.: Contrasting size-resolved hygroscopicity of fine particles derived by HTDMA and HR-ToF-AMS measurements between summer and winter in Beijing: the impacts of aerosol aging and local emissions, Atmospheric Chemistry and Physics, 20, 915-929, 10.5194/acp-20-915-2020, 2020.

Gao, C. Y., Bauer, S. E., Tsigaridis, K., and Im, U.: Global Influence of Organic Aerosol Volatility on Aerosol Microphysical Processes: Composition and Number, Journal of Advances in Modeling Earth Systems, 16, 10.1029/2023ms004185, 2024.

Gui, K., Yao, W., Che, H., An, L., Zheng, Y., Li, L., Zhao, H., Zhang, L., Zhong, J., Wang, Y., and Zhang, X.: Record-breaking dust loading during two mega dust

storm events over northern China in March 2021: aerosol optical and radiative properties and meteorological drivers, Atmospheric Chemistry and Physics, 22, 7905-7932, 10.5194/acp-22-7905-2022, 2022.

Gui, K., Che, H., Yao, W., Zheng, Y., Li, L., An, L., Wang, H., Wang, Y., Wang, Z., Ren, H.-L., Sun, J., Li, J., and Zhang, X.: Quantifying the contribution of local drivers to observed weakening of spring dust storm frequency over northern China (1982–2017), Science of The Total Environment, 894, 10.1016/j.scitotenv.2023.164923, 2023.

Gunthe, S. S., King, S. M., Rose, D., Chen, Q., Roldin, P., Farmer, D. K., Jimenez, J. L., Artaxo, P., Andreae, M. O., Martin, S. T., and Pöschl, U.: Cloud condensation nuclei in pristine tropical rainforest air of Amazonia: size-resolved measurements and modeling of atmospheric aerosol composition and CCN activity, Atmospheric Chemistry and Physics, 9, 7551-7575, DOI 10.5194/acp-9-7551-2009, 2009.

Hu, X., Sun, J., Xia, C., Shen, X., Zhang, Y., Zhang, X., and Zhang, S.: Simultaneous measurements of PM1 and PM10 aerosol scattering properties and their relationships in urban Beijing: A two-year observation, Science of The Total Environment, 770, 10.1016/j.scitotenv.2021.145215, 2021.

Huffman, J. A., Ziemann, P. J., Jayne, J. T., Worsnop, D. R., and Jimenez, J. L.: Development and Characterization of a Fast-Stepping/Scanning Thermodenuder for Chemically-Resolved Aerosol Volatility Measurements, Aerosol Science and Technology, 42, 395-407, 10.1080/02786820802104981, 2008.

Jiang, S., Ye, X., Wang, R., Tao, Y., Ma, Z., Yang, X., and Chen, J.: Measurements of nonvolatile size distribution and its link to traffic soot in urban Shanghai, Science of The Total Environment, 615, 452-461, 10.1016/j.scitotenv.2017.09.176, 2018.

Johnson, G. R., Ristovski, Z., and Morawska, L.: Method for measuring the hygroscopic behaviour of lower volatility fractions in an internally mixed aerosol, J Aerosol Sci, 35, 443-455, 10.1016/j.jaerosci.2003.10.008, 2004.

Kaaden, N., Massling, A., Schladitz, A., Müller, T., Kandler, K., Schütz, L., Weinzierl, B., Petzold, A., Tesche, M., Leinert, S., Deutscher, C., Ebert, M., Weinbruch, S., and Wiedensohler, A.: State of mixing, shape factor, number size distribution, and hygroscopic growth of the Saharan anthropogenic and mineral dust aerosol at Tinfou, Morocco, Tellus B: Chemical and Physical Meteorology, 61, 10.1111/j.1600-0889.2008.00388.x, 2009.

Kim, J.-S. and Park, K.: Atmospheric Aging of Asian Dust Particles During Long Range Transport, Aerosol Science and Technology, 46, 913-924, 10.1080/02786826.2012.680984, 2012.

Kim, J.-S., Kim, Y. J., and Park, K.: Measurements of hygroscopicity and volatility of atmospheric ultrafine particles in the rural Pearl River Delta area of China, Atmospheric Environment, 45, 4661-4670, 10.1016/j.atmosenv.2011.05.054, 2011.

Kurai, J., Watanabe, M., Tomita, K., Yamasaki, H. S. A., and Shimizu, E.: Influence of Asian Dust Particles on Immune Adjuvant Effects and Airway Inflammation in Asthma Model Mice, PLoS ONE, 9, 10.1371/journal.pone.0111831, 2014.

Lei, L., Zhou, W., Chen, C., He, Y., Li, Z., Sun, J., Tang, X., Fu, P., Wang, Z., and Sun,

Y.: Long-term characterization of aerosol chemistry in cold season from 2013 to 2020 in Beijing, China, Environmental Pollution, 268, 10.1016/j.envpol.2020.115952, 2021.

Li, X., Chen, Y., Li, Y., Cai, R., Li, Y., Deng, C., Wu, J., Yan, C., Cheng, H., Liu, Y., Kulmala, M., Hao, J., Smith, J. N., and Jiang, J.: Seasonal variations in composition and sources of atmospheric ultrafine particles in urban Beijing based on near-continuous measurements, Atmospheric Chemistry and Physics, 23, 14801-14812, 10.5194/acp-23-14801-2023, 2023.

Liang, Y., Che, H., Wang, H., Zhang, W., Li, L., Zheng, Y., Gui, K., Zhang, P., and Zhang, X.: Aerosols Direct Radiative Effects Combined Ground-Based Lidar and Sun-Photometer Observations: Cases Comparison between Haze and Dust Events in Beijing, Remote Sensing, 14, 10.3390/rs14020266, 2022.

Liu, J., Zhang, F., Ren, J., Chen, L., Zhang, A., Wang, Z., Zou, S., Xu, H., and Yue, X.: The evolution of aerosol mixing state derived from a field campaign in Beijing: implications for particle aging timescales in urban atmospheres, Atmospheric Chemistry and Physics, 25, 5075-5086, 10.5194/acp-25-5075-2025, 2025.

Lu, J., Shen, X., Ma, Q., Yu, A., Hu, X., Zhang, Y., Liu, Q., Liu, S., Che, H., Zhang, X., and Sun, J.: Size-resolved effective density of ambient aerosols measured by an AAC–SMPS tandem system in Beijing, Atmospheric Environment, 318, 10.1016/j.atmosenv.2023.120226, 2024.

Massling, A., Leinert, S., Wiedensohler, A., and Covert, D.: Hygroscopic growth of sub-micrometer and one-micrometer aerosol particles measured during ACE-Asia, Atmospheric Chemistry and Physics, 7, 3249-3259, DOI 10.5194/acp-7-3249-2007, 2007.

Massoli, P., Lambe, A. T., Ahern, A. T., Williams, L. R., Ehn, M., Mikkilä, J., Canagaratna, M. R., Brune, W. H., Onasch, T. B., Jayne, J. T., Petäjä, T., Kulmala, M., Laaksonen, A., Kolb, C. E., Davidovits, P., and Worsnop, D. R.: Relationship between aerosol oxidation level and hygroscopic properties of laboratory generated secondary organic aerosol (SOA) particles, Geophysical Research Letters, 37, 10.1029/2010gl045258, 2010.

McFiggans, G., Artaxo, P., Baltensperger, U., Coe, H., Facchini, M. C., Feingold, G., Fuzzi, S., Gysel, M., Laaksonen, A., Lohmann, U., Mentel, T. F., Murphy, D. M., O'Dowd, C. D., Snider, J. R., and Weingartner, E.: The effect of physical and chemical aerosol properties on warm cloud droplet activation, Atmospheric Chemistry and Physics, 6, 2593-2649, DOI 10.5194/acp-6-2593-2006, 2006.

Pan, X. L., Yan, P., Tang, J., Ma, J. Z., Wang, Z. F., Gbaguidi, A., and Sun, Y. L.: Observational study of influence of aerosol hygroscopic growth on scattering coefficient over rural area near Beijing mega-city, Atmospheric Chemistry and Physics, 9, 7519-7530, DOI 10.5194/acp-9-7519-2009, 2009.

Pöhlker, M. L., Pöhlker, C., Quaas, J., Mülmenstädt, J., Pozzer, A., Andreae, M. O., Artaxo, P., Block, K., Coe, H., Ervens, B., Gallimore, P., Gaston, C. J., Gunthe, S. S., Henning, S., Herrmann, H., Krüger, O. O., McFiggans, G., Poulain, L., Raj, S. S., Reyes-Villegas, E., Royer, H. M., Walter, D., Wang, Y., and Pöschl, U.: Global organic and inorganic aerosol hygroscopicity and its effect on radiative forcing,

Nature Communications, 14, 10.1038/s41467-023-41695-8, 2023.

Pringle, K. J., Tost, H., Pozzer, A., Pöschl, U., and Lelieveld, J.: Global distribution of the effective aerosol hygroscopicity parameter for CCN activation, Atmospheric Chemistry and Physics, 10, 5241-5255, 10.5194/acp-10-5241-2010, 2010.

Rissler, J., Svenningsson, B., Fors, E. O., Bilde, M., and Swietlicki, E.: An evaluation and comparison of cloud condensation nucleus activity models: Predicting particle critical saturation from growth at subsaturation, Journal of Geophysical Research: Atmospheres, 115, 10.1029/2010jd014391, 2010.

Rose, D., Nowak, A., Achtert, P., Wiedensohler, A., Hu, M., Shao, M., Zhang, Y., Andreae, M. O., and Pöschl, U.: Cloud condensation nuclei in polluted air and biomass burning smoke near the mega-city Guangzhou, China - Part 1: Size-resolved measurements and implications for the modeling of aerosol particle hygroscopicity and CCN activity, Atmospheric Chemistry and Physics, 10, 3365-3383, DOI 10.5194/acp-10-3365-2010, 2010.

Schladitz, A., Müller, T., Nowak, A., Kandler, K., Lieke, K., Massling, A., and Wiedensohler, A.: In situ aerosol characterization at Cape Verde: Part 1: Particle number size distributions, hygroscopic growth and state of mixing of the marine and Saharan dust aerosol, Tellus B: Chemical and Physical Meteorology, 63, 10.1111/j.1600-0889.2011.00569.x, 2011.

Schladitz, A., Müller, T., Kaaden, N., Massling, A., Kandler, K., Ebert, M., Weinbruch, S., Deutscher, C., and Wiedensohler, A.: In situ measurements of optical properties at Tinfou (Morocco) during the Saharan Mineral Dust Experiment SAMUM 2006, Tellus B: Chemical and Physical Meteorology, 61, 10.1111/j.1600-0889.2008.00397.x, 2009.

Shen, X., Sun, J., Che, H., Zhang, Y., Zhou, C., Gui, K., Xu, W., Liu, Q., Zhong, J., Xia, C., Hu, X., Zhang, S., Wang, J., Liu, S., Lu, J., Yu, A., and Zhang, X.: Characterization of dust-related new particle formation events based on long-term measurement in the North China Plain, Atmospheric Chemistry and Physics, 23, 8241-8257, 10.5194/acp-23-8241-2023, 2023.

Shi, J., Hong, J., Ma, N., Luo, Q., He, Y., Xu, H., Tan, H., Wang, Q., Tao, J., Zhou, Y., Han, S., Peng, L., Xie, L., Zhou, G., Xu, W., Sun, Y., Cheng, Y., and Su, H.: Measurement report: On the difference in aerosol hygroscopicity between high and low relative humidity conditions in the North China Plain, Atmospheric Chemistry and Physics, 22, 4599-4613, 10.5194/acp-22-4599-2022, 2022.

Song, X., Wang, Y., Huang, X., Wang, Y., Li, Z., Zhu, B., Ren, R., An, J., Yan, J., Zhang, R., Shang, Y., and Zhan, P.: The Impacts of Dust Storms With Different Transport Pathways on Aerosol Chemical Compositions and Optical Hygroscopicity of Fine Particles in the Yangtze River Delta, Journal of Geophysical Research: Atmospheres, 128, 10.1029/2023jd039679, 2023.

Sun, J., Wang, Z., Zhou, W., Xie, C., Wu, C., Chen, C., Han, T., Wang, Q., Li, Z., Li, J., Fu, P., Wang, Z., and Sun, Y.: Measurement report: Long-term changes in black carbon and aerosol optical properties from 2012 to 2020 in Beijing, China, Atmospheric Chemistry and Physics, 22, 561-575, 10.5194/acp-22-561-2022, 2022.

Tang, M., Chan, C. K., Li, Y. J., Su, H., Ma, Q., Wu, Z., Zhang, G., Wang, Z., Ge, M., Hu, M., He, H., and Wang, X.: A review of experimental techniques for aerosol hygroscopicity studies, Atmospheric Chemistry and Physics, 19, 12631-12686, 10.5194/acp-19-12631-2019, 2019.

Wang, Y., Wu, Z., Ma, N., Wu, Y., Zeng, L., Zhao, C., and Wiedensohler, A.: Statistical analysis and parameterization of the hygroscopic growth of the sub-micrometer urban background aerosol in Beijing, Atmospheric Environment, 175, 184-191, 10.1016/j.atmosenv.2017.12.003, 2018.

Wang, Y., Zhang, F., Li, Z., Tan, H., Xu, H., Ren, J., Zhao, J., Du, W., and Sun, Y.: Enhanced hydrophobicity and volatility of submicron aerosols under severe emission control conditions in Beijing, Atmospheric Chemistry and Physics, 17, 5239-5251, 10.5194/acp-17-5239-2017, 2017.

Wang, Y., Li, Z., Zhang, R., Jin, X., Xu, W., Fan, X., Wu, H., Zhang, F., Sun, Y., Wang, Q., Cribb, M., and Hu, D.: Distinct Ultrafine- and Accumulation-Mode Particle Properties in Clean and Polluted Urban Environments, Geophysical Research Letters, 46, 10918-10925, 10.1029/2019gl084047, 2019.

Wang, Z. B., Hu, M., Wu, Z. J., Yue, D. L., He, L. Y., Huang, X. F., Liu, X. G., and Wiedensohler, A.: Long-term measurements of particle number size distributions and the relationships with air mass history and source apportionment in the summer of Beijing, Atmospheric Chemistry and Physics, 13, 10159-10170, 10.5194/acp-13-10159-2013, 2013.

Wu, Z. J., Zheng, J., Shang, D. J., Du, Z. F., Wu, Y. S., Zeng, L. M., Wiedensohler, A., and Hu, M.: Particle hygroscopicity and its link to chemical composition in the urban atmosphere of Beijing, China, during summertime, Atmospheric Chemistry and Physics, 16, 1123-1138, 10.5194/acp-16-1123-2016, 2016.

Wu, Z. J., Ma, N., Größ, J., Kecorius, S., Lu, K. D., Shang, D. J., Wang, Y., Wu, Y. S., Zeng, L. M., Hu, M., Wiedensohler, A., and Zhang, Y. H.: Thermodynamic properties of nanoparticles during new particle formation events in the atmosphere of North China Plain, Atmospheric Research, 188, 55-63, 10.1016/j.atmosres.2017.01.007, 2017.

Xia, C., Sun, J., Qi, X., Shen, X., Zhong, J., Zhang, X., Wang, Y., Zhang, Y., and Hu, X.: Observational study of aerosol hygroscopic growth on scattering coefficient in Beijing: A case study in March of 2018, Science of The Total Environment, 685, 239-247, 10.1016/j.scitotenv.2019.05.283, 2019.

Xu, W., Xie, C., Karnezi, E., Zhang, Q., Wang, J., Pandis, S. N., Ge, X., Zhang, J., An, J., Wang, Q., Zhao, J., Du, W., Qiu, Y., Zhou, W., He, Y., Li, Y., Li, J., Fu, P., Wang, Z., Worsnop, D. R., and Sun, Y.: Summertime aerosol volatility measurements in Beijing, China, Atmospheric Chemistry and Physics, 19, 10205-10216, 10.5194/acp-19-10205-2019, 2019.

Xu, W. Q., Sun, Y. L., Chen, C., Du, W., Han, T. T., Wang, Q. Q., Fu, P. Q., Wang, Z. F., Zhao, X. J., Zhou, L. B., Ji, D. S., Wang, P. C., and Worsnop, D. R.: Aerosol composition, oxidation properties, and sources in Beijing: results from the 2014 Asia-Pacific Economic Cooperation summit study, Atmospheric Chemistry and

Physics, 15, 13681-13698, 10.5194/acp-15-13681-2015, 2015.

Yin, D., Song, Q., Guo, Y., Jiang, Y., Dong, Z., Zhao, B., Wang, S., Gao, D., Chang, X., Zheng, H., Li, S., Li, Y., and Liu, B.: Regional transport characteristics of PM2.5 pollution events in Beijing during 2018–2021, Journal of Environmental Sciences, 152, 503-515, 10.1016/j.jes.2024.05.044, 2025.

Yu, A., Lu, J., Shen, X., Hu, X., Zhang, Y., Liu, Q., Tong, H., Liang, L., Liu, L., Ma, Q., Han, L., Che, H., Zhang, X., and Sun, J.: Determination of the deposition of urban submicron aerosols in the human respiratory tract considering hygroscopic growth, Atmospheric Environment, 356, 10.1016/j.atmosenv.2025.121289, 2025a.

Yu, A., Shen, X., Ma, Q., Lu, J., Hu, X., Zhang, Y., Liu, Q., Liang, L., Liu, L., Liu, S., Tong, H., Che, H., Zhang, X., and Sun, J.: Size-resolved hygroscopicity and volatility properties of ambient urban aerosol particles measured by a volatility hygroscopicity tandem differential mobility analyzer system in Beijing, Atmospheric Chemistry and Physics, 25, 3389-3412, 10.5194/acp-25-3389-2025, 2025b.

Zhang, S., Shen, X., Sun, J., Zhang, Y., Zhang, X., Xia, C., Hu, X., Zhong, J., Wang, J., and Liu, S.: Atmospheric Particle Hygroscopicity and the Influence by Oxidation State of Organic Aerosols in Urban Beijing, Journal of Environmental Sciences, 124, 544-556, 10.1016/j.jes.2021.11.019, 2023a.

Zhang, S., Shen, X., Sun, J., Che, H., Zhang, Y., Liu, Q., Xia, C., Hu, X., Zhong, J., Wang, J., Liu, S., Lu, J., Yu, A., and Zhang, X.: Seasonal variation of particle hygroscopicity and its impact on cloud-condensation nucleus activation in the Beijing urban area, Atmospheric Environment, 302, 10.1016/j.atmosenv.2023.119728, 2023b.

Zhang, S. L., Ma, N., Kecorius, S., Wang, P. C., Hu, M., Wang, Z. B., Größ, J., Wu, Z. J., and Wiedensohler, A.: Mixing state of atmospheric particles over the North China Plain, Atmospheric Environment, 125, 152-164, 10.1016/j.atmosenv.2015.10.053, 2016.

Zhong, J., Zhang, X., Wang, Y., Sun, J., Shen, X., Xia, C., and Zhang, W.: Attribution of the worse aerosol pollution in March 2018 in Beijing to meteorological variability, Atmospheric Research, 250, 10.1016/j.atmosres.2020.105294, 2021.

Zhong, J., Zhang, X., Dong, Y., Wang, Y., Liu, C., Wang, J., Zhang, Y., and Che, H.: Feedback effects of boundary-layer meteorological factors on cumulative explosive growth of PM2.5 during winter heavy pollution episodes in Beijing from 2013 to 2016, Atmospheric Chemistry and Physics, 18, 247-258, 10.5194/acp-18-247-2018, 2018.

---

## Author Comment (AC3)

**Response to Reviewer #2**

This manuscript describes size distribution, volatility and hygroscopicity measurements taken during a 1-month campaign in Beijing. The analysis focuses on the hygroscopic growth factor at 90% RH, the volatility shrink factor at 300℃, and the number fractions or more hygroscopic/more volatile particles. Auxiliary data include PM2.5 and PM10 from a monitoring station.

Overall this measurement report paper is easy to read. The measurement methodology appears to be sound. Data for each figure are available in a publicly available repository. The conclusions are limited to descriptive statistics of the data. Per journal guidelines, measurement reports should present substantial new measurement results. Manuscripts may be considered for publication even if broader implications for atmospheric chemistry and physics may be less developed. In the opinion of this referee, this manuscript at the cusp of the "substantial new measurement results" threshold. However, there are concerns about the broader utility of the dataset, likely limiting its overall impact on the field.

**Reply:** Thanks for your comments. Understanding the aerosol hygroscopicity and volatility in different conditions is crucial for determining their environment and climate effects. The understanding of characteristics of hygroscopicity and volatility under different polluted conditions remains inadequately understood. Simultaneous measurements of aerosol hygroscopicity and volatility were performed using Volatility-Hygroscopicity Tandem Differential Mobility Analyzer during April 2024 in Beijing. The study aimed to enhance understanding aerosol mixing state and evolution under different conditions and provide reliable observational constraints for reducing discrepancies between simulation results and observational data. We restructured the manuscript and added more comparison between the results in this study and previous researches, hopefully, it meets the requirements of ACP.

**Major comments**

The novelty/utility of this dataset is not entirely clear. The data are novel in the sense that they have not been published before and add to the available datasets on the topic. However, size-resolved hygroscopicity and to a lesser extent volatility datasets have been widely available for more than a decade. No clear new conclusions that "substantial advances and general implications for the scientific understanding of atmospheric

chemistry" could be drawn from the work (hence the measurement report and focusing on descriptive results). But even as a measurement report, the utility of the dataset seems limited without additional contextualizing data. It is not immediately clear how this data could be used in future studies to advance the field further.

**Reply:** Thanks for reviewer's comments. We have reorganized the introduction and reviewed the previous researches in the introduction. As reviewer's mentioned, aerosol hygroscopicity or volatility have been widely investigated worldwide. However, the simultaneous study of hygroscopicity and volatility in China is still limited (Cai et al., 2017; Kim et al., 2011; Wang et al., 2017; Yu et al., 2025; Zhang et al., 2016), especially under different pollution environments. Besides, one of the most important natural aerosols in the atmosphere, dust aerosols significantly affect atmospheric chemistry, human health, climate change, and biogeochemical cycles (Chen et al., 2021; Kurai et al., 2014). Schladitz et al. (2011) demonstrated that the influence of dust particles was observed down to 300 nm during the Saharan Mineral Dust Experiment (SAMUM). Previous studies revealed that the changes of submicron aerosol effective density and optical properties during the dust period (Lu et al., 2024; Xia et al., 2019). Lu et al. (2024) found that the effective densities of 150, 250, 350, 450 nm under dusty conditions were higher than those during non-dusty periods, which reflected the dust influence on accumulation mode particles. Although the climatic and environmental effects of dust are considerable, limited studies focus on the dust effect on aerosol hygroscopicity and volatility simultaneously, especially on submicron aerosols. This study aims to enhance understanding aerosol mixing state and evolution under different conditions and offer reliable observational constraints for reducing discrepancies between simulation results and observational data. We have restructured the manuscript and strengthened the comparison with existing researches. Besides, we added the analysis with aerosol optical properties' data to explore the dust effect on aerosol properties. The results of aerosol hygroscopicity and volatility under different pollution environments and a case study of a dust event advance our understanding of aerosol mixing state and evolution under different conditions, providing reliable observational constraints for model evaluation.

The analysis of the dust event should be omitted from the title abstract and conclusion. Yes, PM$_{2.5}$ and PM$_{10}$ increase for a few hours due to dust. But there are three significant weaknesses. First, only a single dust event is presented. Second, the dust event comes with a different airmass, which unsurprisingly has different overall aerosol properties. These

may or may not be related to dust. Finally, it is unclear how the mostly supermicron dust is relevant for the sizes for HGF and VSF (50 – 300 nm). Perhaps some fraction of particles in the 300 nm size bin are dust, but that fraction is unclear. At minimum the authors should show that aerosol volume from the size distribution (total and 250-350 nm) correlates with the increase $PM_{2.5}$.

**Reply:** Thanks for reviewer's suggestions. We have changed the title of this manuscript and reorganized the content. We focus on the characteristics of hygroscopicity and volatility of submicron aerosols under different pollution environment, including this dust event. Although only one dust event is presented, it represents a typical spring dust episode in Beijing—an important atmospheric process that contributes significantly to regional particulate pollution, and limited researches focused on the dust effect on size-resolved aerosol hygroscopicity and volatility. Previous studies showed that Mongolia is an important source of dust for China (Chen et al., 2023; Zhang et al., 2024). The back trajectories analysis during the dust period were displayed in the supplement (Figure. S2), which showed that the air mass containing dust particles mainly originated from the central and western of Mongolia. The figure R1 showed the variation of particle volume size distribution, surface size distribution and number size distribution on April 15, 2024. During the dust period (6:00-12:00), accumulation mode particle number decreased, while the accumulation mode particle surface and volume remained large. Figure R2 showed variation of particle volume fraction size distribution, surface fraction size distribution and number fraction size distribution on April 15, 2024. Volume and surface fraction were large during the dust period, implying the dust effect on accumulation mode particles. Dust particles, suspended in the atmosphere, range from less than 0.1µm to over 100µm (Adebiyi et al., 2023). Previous studies have observed dust particles in submicron aerosols (Hu et al., 2012; Panta et al., 2023). Although we didn't have the aerosol chemical composition data, we used aerosol optical properties, size-resolved aerosol hygroscopicity and volatility to demonstrated that dust particles effect on submicron aerosols. Low SAE for $PM_1$ and $PM_{10}$ suggested the dominated aerosols were affected by dust. The mean f(80%) for $PM_1$ and $PM_{10}$ was 1.03 and 1.02 during the dust period. The results of size-resolved hygroscopicity and volatility showed high number fraction of non-volatile components and nearly hydrophobic particles at 200 and 300 nm during the dust period suggested that dust particles can be as small as 200 nm in diameter. We have revised this manuscript thoroughly. This analysis not only enriches the dataset by

capturing aerosol properties under dust conditions but also provides critical evidence into how dust events modulate submicron aerosol physicochemical properties.

[Figure]

Figure R1. The variation of particle volume size distribution, surface size distribution and number size distribution on April 15, 2024.

[Figure]

Figure R2. The variation of particle volume fraction size distribution, surface fraction size distribution and number fraction size distribution on April 15, 2024.

**Reference:**

Cai, M., Tan, H., Chan, C. K., Mochida, M., Hatakeyama, S., Kondo, Y., Schurman, M. I., Xu, H., Li, F., Shimada, K., Li, L., Deng, Y., Yai, H., Matsuki, A., Qin, Y., and Zhao, J.: Comparison of Aerosol Hygroscopcity, Volatility, and Chemical Composition between a Suburban Site in the Pearl River Delta Region and a Marine Site in Okinawa, Aerosol and Air Quality Research, 17, 3194-3208, 10.4209/aaqr.2017.01.0020, 2017.

Hu, M., Peng, J., Sun, K., Yue, D., Guo, S., Wiedensohler, A., and Wu, Z.: Estimation of Size-Resolved Ambient Particle Density Based on the Measurement of Aerosol Number, Mass, and Chemical Size Distributions in the Winter in Beijing, Environmental Science & Technology, 46, 9941-9947, 10.1021/es204073t, 2012.

Kim, J.-S., Kim, Y. J., and Park, K.: Measurements of hygroscopicity and volatility of atmospheric ultrafine particles in the rural Pearl River Delta area of China, Atmospheric Environment, 45, 4661-4670, 10.1016/j.atmosenv.2011.05.054, 2011.

Panta, A., Kandler, K., Alastuey, A., González-Flórez, C., González-Romero, A., Klose, M., Querol, X., Reche, C., Yus-Díez, J., and Pérez García-Pando, C.: Insights into the single-particle composition, size, mixing state, and aspect ratio of freshly emitted mineral dust from field measurements in the Moroccan Sahara using electron microscopy, Atmospheric Chemistry and Physics, 23, 3861-3885, 10.5194/acp-23-3861-2023, 2023.

Wang, Y., Zhang, F., Li, Z., Tan, H., Xu, H., Ren, J., Zhao, J., Du, W., and Sun, Y.: Enhanced hydrophobicity and volatility of submicron aerosols under severe emission control conditions in Beijing, Atmospheric Chemistry and Physics, 17, 5239-5251, 10.5194/acp-17-5239-2017, 2017.

Yu, A., Lu, J., Shen, X., Hu, X., Zhang, Y., Liu, Q., Tong, H., Liang, L., Liu, L., Ma, Q., Han, L., Che, H., Zhang, X., and Sun, J.: Determination of the deposition of urban submicron aerosols in the human respiratory tract considering hygroscopic growth, Atmospheric Environment, 356, 10.1016/j.atmosenv.2025.121289, 2025.

Zhang, S. L., Ma, N., Kecorius, S., Wang, P. C., Hu, M., Wang, Z. B., Größ, J., Wu, Z. J., and Wiedensohler, A.: Mixing state of atmospheric particles over the North China Plain, Atmospheric Environment, 125, 152-164, 10.1016/j.atmosenv.2015.10.053, 2016.